# Paradoxical imbalance between activated lymphocyte protein synthesis capacity and rapid division rate

Mina O Seedhom[1], Devin Dersh[1], Jaroslav Holly[1], Mariana Pavon-Eternod[1], Jiajie Wei[1], Matthew Angel[1], Lucas Shores[1], Alexandre David[2], Jefferson Santos[1], Heather Hickman[1], Jonathan W Yewdell[1]*

[1]National Institute of Allergy and Infectious Diseases, Bethesda, United States; [2]CNRS UMR-5203; INSERM U661; UM1; UM2, Institut de Génomique Fonctionnelle, Montpellier, France

*For correspondence: jyewdell@nih.gov

Competing interest: The authors declare that no competing interests exist.

**Abstract** Rapid lymphocyte cell division places enormous demands on the protein synthesis machinery. Flow cytometric measurement of puromycylated ribosome-associated nascent chains after treating cells or mice with translation initiation inhibitors reveals that ribosomes in resting lymphocytes in vitro and in vivo elongate at typical rates for mammalian cells. Intriguingly, elongation rates can be increased up to 30% by activation in vivo or fever temperature in vitro. Resting and activated lymphocytes possess abundant monosome populations, most of which actively translate in vivo, while in vitro, nearly all can be stalled prior to activation. Quantitating lymphocyte protein mass and ribosome count reveals a paradoxically high ratio of cellular protein to ribosomes insufficient to support their rapid in vivo division, suggesting that the activated lymphocyte proteome in vivo may be generated in an unusual manner. Our findings demonstrate the importance of a global understanding of protein synthesis in lymphocytes and other rapidly dividing immune cells.

## eLife assessment

This study addresses how protein synthesis in activated lymphocytes keeps up with their rapid division, with **important** findings that are of significance to cell biologists and immunologists endeavouring to understand the 'economy' of the immune system. The work is supported by **solid** data. Because it proposes non-conventional mechanisms, the study sets the scene for further work in this area.

## Introduction

Naive lymphocytes are among the smallest nucleated cells in mammals – nearly devoid of cytoplasm, with few mitochondria – and have minimal metabolic activity, consistent with doubling times on the order of hundreds to thousands of days, respectively, for B cells (*Macallan et al., 2005*; *van Gent et al., 2008*) and T cells (*Vrisekoop et al., 2008*). Within a day of activation by cognate antigen, lymphocytes begin to divide rapidly, with reported doubling times as rapid as 6 hr (*Jelley-Gibbs et al., 2000*; *Zhang et al., 1988*). Such Jekyll and Hyde behavior requires massive induction of DNA and protein synthesis to support daughter cell production (*Marshall and Roberts, 1963*) as well as synthesizing large amounts of immune regulatory (e.g., cytokines) and effector molecules (e.g., antibody and cytokines) (*Ripps and Hirschhorn, 1967*; *Gery and Waksman, 1972*; *Mier and Gallo, 1980*).

Pioneering studies of protein synthesis regulation in lymphocytes utilizing radiolabeled amino acids on mitogen-activated human peripheral blood lymphocytes reported 7- to 20-fold increases in protein synthesis activity (*Hirschhorn et al., 1963*; *Kay and Korner, 1966*). While this is an impressive increase, it was assumed that sufficient protein was synthesized to enable the generation of daughter cells with the same protein content as their progenitor. Moreover, radiolabeling, like all methods, is imperfect, and its accuracy as a measure of protein synthesis rates depends on assumptions that are nearly impossible to definitively verify (*Yewdell et al., 2011*). Applying new methods to old problems is a tried-and-true method for generating new insights and discoveries.

Indeed, newer methods, including ribosome profiling (*Ingolia et al., 2011*), tRNA arrays (*Dittmar et al., 2006*), and tandem mass spectrometry (*Schwanhäusser et al., 2009*), are revolutionizing the field of protein synthesis. This includes extending classical methods. Puromycin (PMY) is an aminonucleoside antibiotic that mimics tyrosine-tRNA, binding the ribosome A site and causing rapid chain termination by covalently attaching to the C-terminus of the nascent chain. PMY was first applied in classical protein synthesis studies (*Pestka, 1971*) and remains a workhorse in understanding ribosomal catalysis of protein synthesis (*Eggers et al., 1997*; *Schmidt et al., 2009*; *Liu et al., 2012*; *Nakano and Hara, 1979*).

We developed the ribopuromycylation method (RPM) to better localize and quantify active protein synthesis. RPM uses a brief pulse of PMY to label elongating nascent chains frozen on ribosomes by treating cells with a translation elongation inhibitor. Ribosome-bound nascent chains are then detected using a PMY-specific monoclonal antibody in fixed and permeabilized cells via standard immunofluorescence (*David et al., 2012*) or flow cytometry (*Seedhom et al., 2016*).

Here, we use RPM, and the ribosome transit assay (RTA), an extension of RPM that measures elongation rates, in conjunction with classic techniques to quantify the number and protein synthesis activity of ribosomes in resting and activated human and mouse lymphocytes. Our findings reveal novel features of lymphocyte translation as well as a discrepancy in the protein synthesis capacity of T cells with respect to their rapid in vivo division rates, emphasizing the importance of quantitative accounting as a reality check for our limited understanding of fundamental aspects of cell biology and immunology.

## Results

### Characterizing protein synthesis in human lymphocytes ex vivo with flow RPM implicates widespread ribosome stalling in non-activated cells

We first used flow RPM to compare translation in non-activated vs. Phorbol 12-myristate 13-acetate (PMA)/ionomycin/IL-2-activated human lymphocyte subsets after 2 and 5 days in culture (*Figure 1A*). We devolved the total flow RPM signals into T cell (CD4[+], CD8[+]) and B cell (CD19[+]) subsets to follow distinct patterns of protein synthesis in each population (*Figure 1—figure supplement 1A*). Comparing lymphocytes from three donors revealed considerable donor heterogeneity in RPM staining of day 2 activated cells and proliferation of lymphocyte subpopulations.

We performed RPM on peripheral blood mononuclear cells labeled with carboxyfluorescein succinimidyl ester (CFSE) to track cell division by dye dilution (*Figure 1—figure supplement 1B*). On day 2, activated CD8[+] T cells demonstrated a wide range of RPM staining, with nearly all divided cells at day 5 CFSE[low] and RPM[high]. Some divided cells exhibited near baseline RPM signals, however, consistent with their return to a resting state. Interestingly, although non-activated cells did not divide, ~50% demonstrated increased RPM staining.

We noted that the RPM signal in PMA/ionomycin-activated CD8[+] T cells was only two- to fivefold higher than in non-activated cells. This increase is modest compared to the ~15-fold activation-induced increase in protein synthesis in original studies (*Hirschhorn et al., 1963*; *Kay and Korner, 1966*). To examine this discrepancy, we first incubated cells for 15 min with initiation inhibitors (harringtonine, HAR; pactamycin, PA) or elongation inhibitors (emetine, EME; cycloheximide, CHX), followed by RPM staining. Elongation inhibitors had minor effects on RPM of activated or resting cells (*Figure 1B*), as expected due to ribosome retention of nascent chains (*David et al., 2012*). Initiation inhibitors, however, clearly discriminated between resting and activated cells. RPM signal was diminished by up

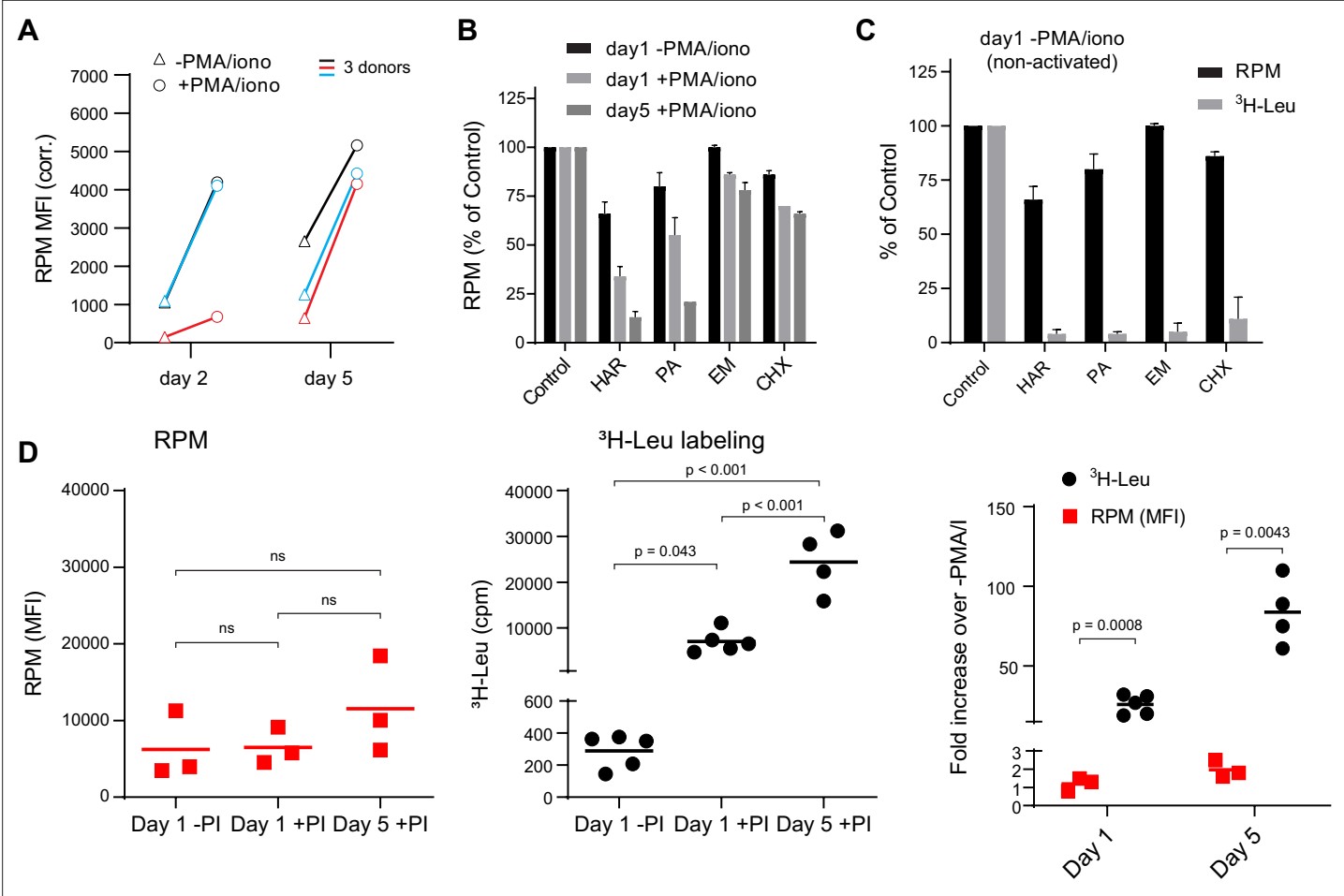

**Figure 1.** Stalled ribosomes in resting ex vivo human lymphocytes. (**A**) Primary human lymphocytes from three independent donors were cultured in PMA/ionomycin and IL-2 (+PMA/iono) or IL-2 only (−PMA/iono) for up to 5 days. CD45+ cells were processed for flow ribopuromycylation method (RPM). (**B**) Primary human lymphocytes were cultured ex vivo as indicated, followed by a 15-min treatment with vehicle, harringtonine (HAR, 5 µg/ml), pactamycin (PA, 10 µM), emetine (EME, 25 µg/ml), or cycloheximide (CHX, 200 µg/ml), and all cultures were then treated with puromycin (PMY, 50 µg/ml) for 5 min. Cells were harvested, and RPM staining was performed. Gated on CD45+ cells. Error bars represent standard deviation of two independent experiments. (**C**) Radioactive amino acid incorporation (0.2 mCi/ml [3H]-Leu for 5 min) or RPM (as in B) in day 1 non-activated human lymphocytes. Error bars represent standard deviation of two independent experiments. (**D**) Radioactive amino acid incorporation and RPM in rested and activated human lymphocytes. RPM MFI values (gated on CD45+ cells) on the left, [3H]-Leu incorporation (cpm) in the middle, and ratios of the activated to the resting cells on the right. Each point represents a single donor; bars indicate the mean from three to five independent donors. Left and middle panels: one-way analysis of variance (ANOVA) pairwise p-values; right panel: unpaired t-test p-values with Welch's correction.

The online version of this article includes the following source data and figure supplement(s) for figure 1:

**Source data 1.** Numerical data and statistics related to *Figure 1*.

**Figure supplement 1.** Ribopuromycylation method (RPM) tracks translation in distinct cell populations over time.

**Figure supplement 1—source data 1.** Numerical data related to *Figure 1—figure supplement 1*.

**Figure supplement 2.** Dominant populations of monosomes in resting human and mouse lymphocytes.

**Figure supplement 2—source data 1.** Numerical data related to *Figure 1—figure supplement 2*.

to 80–90% on day 5 post-activation. Note that at the standard translation rate of 6 amino acids/s, 15 min is sufficient time to complete translation of all but the very longest transcripts.

We repeated this experiment using day 1 resting lymphocytes to directly compare flow RPM with classical metabolic radiolabeling with [3H]-Leucine ([3H]-Leu) (*Figure 1C*). All inhibitors nearly completely blocked incorporation of [3H]-Leu into proteins, suggesting that there were actively translating ribosomes in resting cells and that the inhibitors were active, even though RPM labeling was only weakly impacted. We also performed a time course examining [3H]-Leu incorporation compared

to flow RPM signal in day 1 resting, day 1 activated, and day 5 activated human lymphocytes. Plotting the ratios between activated and non-activated cells from RPM flow vs. [3H]-Leu incorporation revealed a substantial difference between the two methods (*Figure 1D*).

Thus, we cannot attribute the persistence of flow RPM staining in translation initiation inhibitor-treated resting lymphocytes to incomplete inhibition of protein synthesis. Instead, these data are consistent with a significant fraction of 'stalled' ribosomes in cultured resting cells, that is, ribosomes with nascent chains that are not actively translating. Stalled ribosomes would be labeled with PMY, as originally described in neurons (*Graber et al., 2013*), but would not incorporate [3H]-Leu, just as we observe with resting lymphocytes.

## Flow RPM measures ribosome elongation rates in live cells

To extend these findings, we developed a variation of approaches that use initiation inhibitors to measure ribosome transit times, for example by conversion of polysomes to monosomes (*Conn and Qian, 2013*) or ribosome profiling (*Ingolia et al., 2011*). To derive a relative ribosome transit rate, we incubate cells with the initiation inhibitor HAR for increasing times before shifting cells to 4°C to halt ribosome elongation and process for RPM staining (*Figure 2A*).

We validated this approach in HeLa cells whose ribosome transit times are well characterized (*Nielsen and McConkey, 1980*). This revealed a curve that follows one phase exponential decay (*Figure 2B*; gating strategy in *Figure 1—figure supplement 1C*), with a calculated half-life to decay of 70–150 s. Including EME with HAR prevented decay of the RPM signal, as predicted, since EME blocks elongation while enabling (even enhancing) puromycylation (*David et al., 2012*; *David et al., 2013*).

Immunoblotting of puromycylated nascent chains validated the approach by showing a time-dependent decrease in PMY signal and increased $M_r$ of nascent chains after blocking initiation (*Figure 2C*). This is expected since nascent chains present at later time points after blocking initiation will be longer. Incubation of EME with HAR greatly retarded the loss of signal and the shift to longer puromycylated nascent chains.

We applied this RPM-based RTA to investigate translational control in human lymphocytes. Day 5 activated lymphocytes behaved similarly to HeLa cells in their RTA half-life and EME sensitivity (*Figure 2D*). In contrast, in day 1 resting lymphocytes, there was a limited decay in the signal. Furthermore, the decay was similar in EME-treated cells, consistent with the idea that the flow RPM signal in day 1 resting lymphocytes predominantly represents stalled ribosomes with bound nascent chains.

To independently measure ribosome transit times in day 1 resting vs. activated lymphocytes, we treated cells for increasing times with HAR and then pulse labeled with [3H]-Leu (*Figure 2E*). This showed that both resting and activated cells demonstrated a decay half-life of ~90–100 s, similar to the RTA values for activated lymphocytes and HeLa cells.

Based on these findings, we conclude that:

1. A large fraction of ribosomes stalled in resting cultured lymphocytes.
2. Elongation occurs at similar rates for HeLa cells and lymphocytes, with the active ribosomes in resting lymphocytes translating at a similar rate as fully activated lymphocytes.
3. RTA provides a simple flow cytometric measure of ribosome transit rates, confirming and extending the findings of *Argüello et al., 2018* who reported a highly similar method.

## Resting human lymphocytes have a dominant monosome population

Protein synthesis is generally believed to occur predominantly in polysome structures, consisting of multiple ribosomes transiting a single mRNA (*Warner et al., 1963*). Classic (*Cooper et al., 1976*; *Kay et al., 1971*) and more recent studies (*Tan et al., 2017*) have established, however, that resting lymphocytes have few polysomes and provided evidence for active monosome translation by their stability in high salt, which dissociates non-translating ribosomes (*Zylber and Penman, 1970*).

Confirming these reports, we found that a large fraction of assembled ribosomes in resting human lymphocytes fractionate as monosomes in sucrose gradients (*Figure 1—figure supplement 2A*). Polysome abundance increases over 2 days post-activation. Treating freshly isolated human lymphocytes with CHX to freeze ribosomes (*Kay et al., 1971*; *Stanners, 1966*) did not increase polysome recovery (*Figure 1—figure supplement 2B*). These findings, coupled with our RPM/RTA measurements, indicate that stalled ribosomes are likely monosomes.

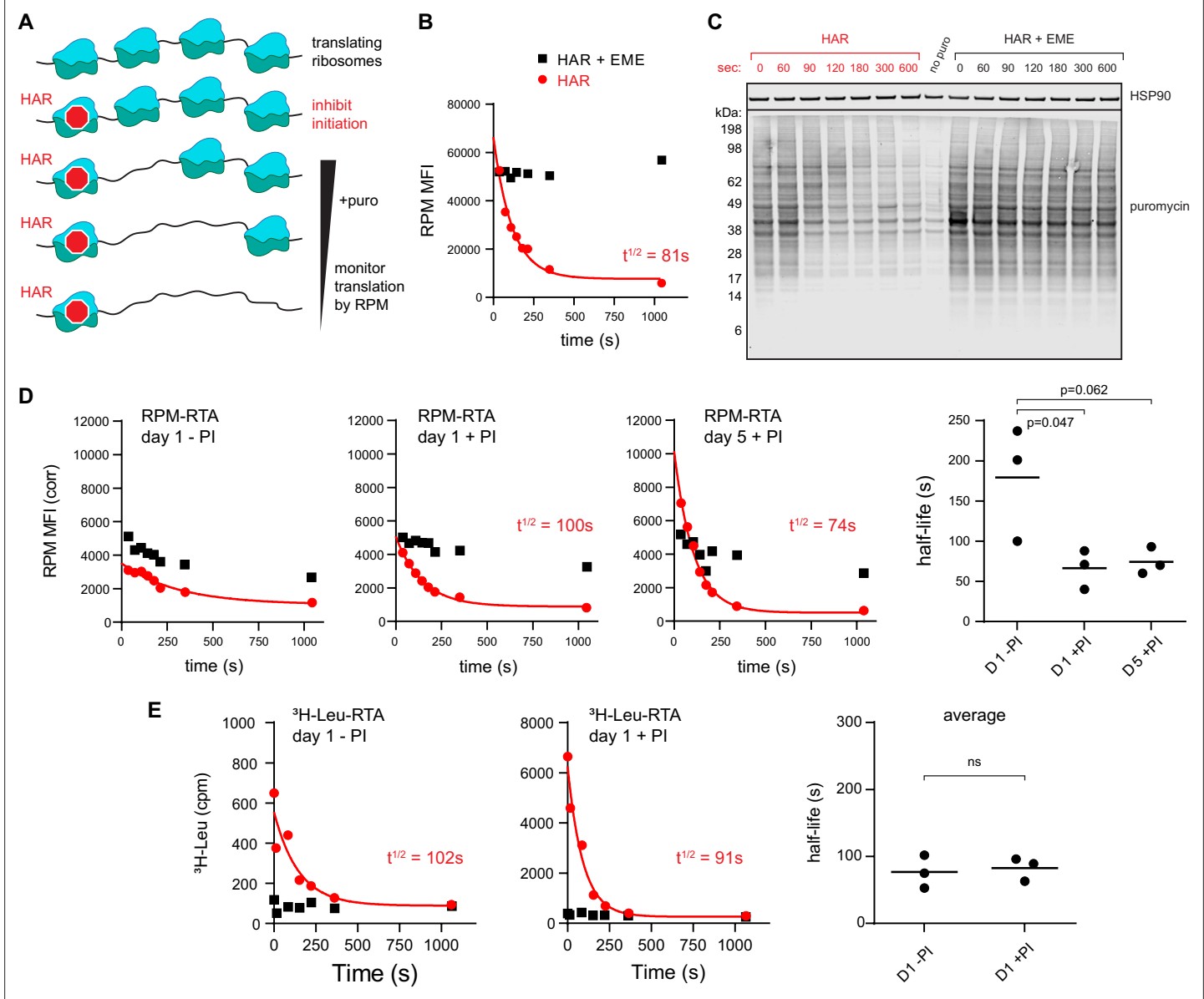

**Figure 2.** Ribopuromycylation method (RPM) measures ribosome transit times in HeLa and human lymphocytes. (**A**) Schematic representation of the RPM ribosome transit analysis (RTA) method. Translation initiation is blocked and the decrease in RPM is monitored as the elongating ribosomes run off mRNA. (**B**) RPM-RTA in HeLa cells. Harringtonine (HAR, 5 µg/ml) is used to inhibit new ribosome initiation; emetine (EME, 25 µg/ml) is used to freeze ribosomes on mRNA; puromycin (PMY, 50 µg/ml) generates RPM signal. Curve is fitted using one phase exponential decay, and ribosome transit times are expressed as RPM half-time to decay. Representative of three biological replicates. (**C**) Same as B, but cells are instead lysed in the presence of MG-132 and subjected to anti-puromycin western blot analysis. Representative of two biological replicates. (**D**) Representative plots of the RPM-RTA signal in resting and activated human lymphocytes (left three panels). Gated on CD45 cells. Far right, ribosome transit times determined from three independent donors. Each dot represents data from one individual donor; the horizontal bars indicate the mean. p-values indicate one-way analysis of variance (ANOVA) pairwise comparisons. (**E**) Ribosome transit times as in A but determined by [³H]-Leu incorporation instead of RPM. After treatment with HAR or HAR plus EME, cells were labeled for 5 min in 0.25 mCi/ml [³H]-Leu. Right panel, ribosome transit times determined by [³H]-Leu incorporation from three independent donors. Each dot represents data from one individual donor; the horizontal bars indicate the mean. Unpaired t-test.

The online version of this article includes the following source data for figure 2:

**Source data 1.** Numerical data and statistics related to *Figure 2*.

**Source data 2.** Uncropped and outlined immunoblot images related to *Figure 2C*.

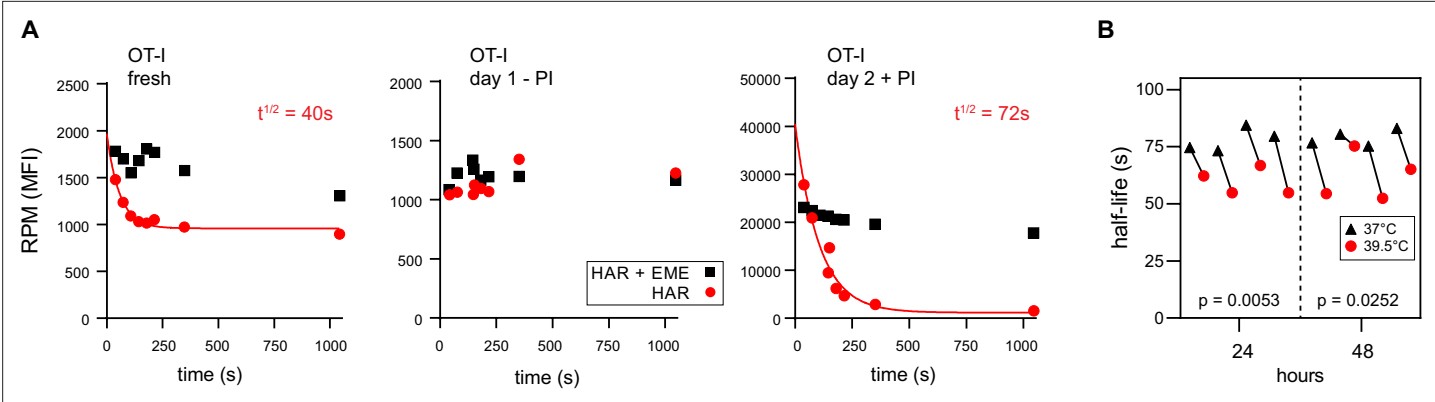

**Figure 3.** Ribopuromycylation method (RPM) ribosome transit analysis of OT-I T cells in vitro. (**A**) Lymphocytes from spleens and lymph nodes from transgenic OT-I mice were isolated, and either used immediately, cultured for 1 day in the absence of PMA/ionomycin, or cultured for 2 days in the presence of PMA/ionomycin and IL-2. RPM-ribosome transit assay (RTA) analysis was conducted to determine ribosome transit half-lives, both with and without emetine (EME). Representative of three biological replicates. (**B**) Lymphocytes from spleens and lymph nodes from transgenic OT-I mice were isolated, labeled with carboxyfluorescein succinimidyl ester (CFSE), and cultured under activating conditions for either 24 or 48 hr. Cells were harvested, and RPM-RTA was performed at both 37 and 39.5°C. Half-life of RPM signal by RTA is plotted; p-values determined by paired t-test analysis. Representative of two biological replicates.

The online version of this article includes the following source data for figure 3:

**Source data 1.** Numerical data and statistics related to *Figure 3*.

## Protein synthesis in mouse lymphocytes ex vivo

Working with human lymphocytes is problematic – preparations between individuals vary considerably, and the manipulations required to isolate lymphocytes from donor blood, such as elutriation and Percoll gradient purification, increase the time cells spend outside their physiological environment.

Seeking a more reproducible system without the impact of potential artefactual stalling of the translation machinery, we turned to OT-I TCR transgenic mice (*Hogquist et al., 1994*). OT-I cells are CD8[+] T cells specific for a defined cognate ligand (mouse K[b] MHC class I molecules bound to the ovalbumin-derived SIINFEKL peptide) that can be activated in vitro or in vivo. OT-I T cells can be obtained from spleen or lymph nodes in reasonable numbers at ~80% purity without further manipulation.

RTA analysis revealed that there was no decay in RPM signal for ex vivo day 1 resting OT-I T cells, consistent with near total stalling of once-translating ribosomes, as we identified in human lymphocytes (*Figure 3A*, middle panel). In contrast, in freshly isolated OT-I cells, the RPM signal decays by 50%, consistent with active translation by 50% of the ribosomes with the rest of the ribosomes likely stalled or poised. The signal decay $t_{1/2}$ of 40 s is consistent with translation of shorter than average mRNAs or stalling on partially translated mRNAs. Polysomes were a minor fraction in freshly isolated mouse lymphocytes, even when mice were pre-treated with CHX (*Figure 1—figure supplement 2C*, further addressed in the next section).

In contrast, day 2 activated ex vivo OT-I T cells demonstrated a 20-fold increased RPM signal relative to resting cells, a near total signal decay with a $t_{1/2}$ of ~70 s (*Figure 3A*, right panel), and a preponderance of polysomes (*Figure 1—figure supplement 2D*). This is consistent with the large fractional engagement of ribosomes upon activation. Notably, the decay rate is faster than observed in previous conditions and intriguingly, the rate increases by ~20% at a 'fever' temperature of 39.5°C (*Figure 3B*). This suggests that lymphocytes may be able to exceed the standard mammalian cell elongation rate of ~6 residues/s (*Fan and Penman, 1970*), particularly under fever conditions, when maximizing T cell protein synthesis is likely at a premium to support their anti-viral activity by rapid division and production of effector molecules.

## Protein synthesis in mouse lymphocytes and innate immune cells in vivo

Mammalian cells evolved, of course, in mammals, not in plastic flasks nurtured by synthetic media in a 20% oxygen atmosphere. We therefore adapted the RTA assay to mice. To simultaneously measure resting and activated T cells, we adoptively transferred CFSE-labeled OT-I T cells into congenic B6

mice, which we infected with SIINFEKL-expressing vaccinia virus (VACV) to activate OT-I cells. We then injected mice with HAR for 0–10 min, followed by PMY injection and flow RPM processing of harvested splenocytes (*Figure 4A*). With each mouse providing a single data point, we could generate RTA curves for non-activated host CD4 and CD8 cells as well as transferred OT-I cells activated by VACV infection (*Figure 4B*). These curves show that nearly all ribosomes with nascent chains in both resting and activated lymphocytes are actively elongating proteins in vivo.

The elongation rate in vivo is surprisingly slower than the in vitro rate. Notably, this experiment used our original protocol of PMY treatment alone (*Seedhom et al., 2016*) since EME, the inhibitor used to stabilize puromycylated polypeptides on ribosomes in vitro (*David et al., 2012*) was ineffective in vivo. We found, however, that CHX is active in vivo, arresting the accumulation of puromycylated polypeptides for at least 60 min after injecting PMY (*Figure 4—figure supplement 1A*). We therefore modified the RTA by simultaneously treating animals with CHX with PMY to determine the relative amount of ribosome-associated nascent chains in vivo. This enabled comparison of translation activity in various immune cell types using 15-min HAR pretreatment values to subtract the signal from stalled ribosomes. The number of translating ribosomes varies over a narrow range among resting splenic lymphocytes, NK cells, macrophages, and neutrophils (*Figure 4—figure supplement 1B*).

Using this improved RPM protocol, 1 day after infecting mice with VACV we now measured a ~15-fold increase in translating ribosomes in activated OT-I T cells in vivo (*Figure 4—figure supplement 1C*; gating strategy in *Figure 4—figure supplement 1D*) as compared to the 10-fold increase we previously reported (*Seedhom et al., 2016*). As cell division progressed over the next 2 days, the signal from translating ribosomes decreased (*Figure 4—figure supplement 1E,F*). Comparing the results from *Figure 4—figure supplement 1C* (OT-I cells) with (polyclonal human CD8[+] T cells) reveals what we had described previously (*Seedhom et al., 2016*), that a transgenic T cell population has much less spread of RPM staining when compared to activated polyclonal T cells in C57/BL6J mice after VACV infection, or here when comparing to all human CD8[+] T cells.

We next performed the modified RTA to measure translation rates in OT-I cells in vivo on day 2 and 3 post-infection with VACV-SIINFEKL (*Figure 4C*). Addition of CHX to the in vivo RTA is important because of the well-characterized 'leakiness' of HAR (*Lee et al., 2012*); indeed, ribosome transit times in activated OT-I cells were now in line with the in vitro rates, and were ~20% faster than transit times in recipient (non-activated) T cells.

These results indicate that:

1. Monitoring accurate translation and rates in vivo is possible and avoids artifacts associated with ex vivo lymphocyte cultures.
2. Activation increases elongation rates in lymphocytes by ~20%.

## Contribution of monosomes vs. polysomes to T cell translation

We next biochemically characterized translation in resting OT-I cells in vivo or OT-I cells activated in vitro by PMA/ionomycin/IL-2. We treated animals/cells with CHX/PMY, isolated ribosomes from cell lysates on sucrose gradients in monosome and polysome fractions, blotted fractions onto nitrocellulose and stained with antibodies against RPL7 or PMY. The robust PMY signal shows that, contrary to recent claims (*Enam et al., 2020*; *Hobson et al., 2020*), PMY does not completely release nascent chains when ribosomes are previously exposed to CHX (*Figure 5A*).

After setting the puromycylation:RPL7 ratio in polysomes to 100% (assuming that all ribosomes in the polysome fraction are actively translating), we found that 33% of monosomes in resting in vivo OT-I T cells and 52% of monosomes in day 2 activated OT-I T cells were puromycylated. Since in vivo RTA indicates that there is essentially no stalling of puromycylated ribosomes (*Figure 4C*), these data demonstrate robust translation in T cell monosomes. Assuming equal elongation rates, ~38% and ~32% of overall translation would occur in monosomes of resting in vivo and activated in vitro cells, respectively. We note, however, that since PMY reduces the number of polysomes recovered from CHX-treated cells by 5 – 10%, a small fraction of translating monosomes probably derive from the polysome population.

The high fraction of monosome-based translation is surprising in activated cells. We noted that the activation protocol for OT-I T cells we used is far less effective than that published by *Tan et al., 2017* which includes SIINFEKL antigenic stimulation along with PHA and ionomycin. Bulk peptide-antigen stimulation directly ex vivo is not possible with human cells, but it is with transgenic murine T cells, and

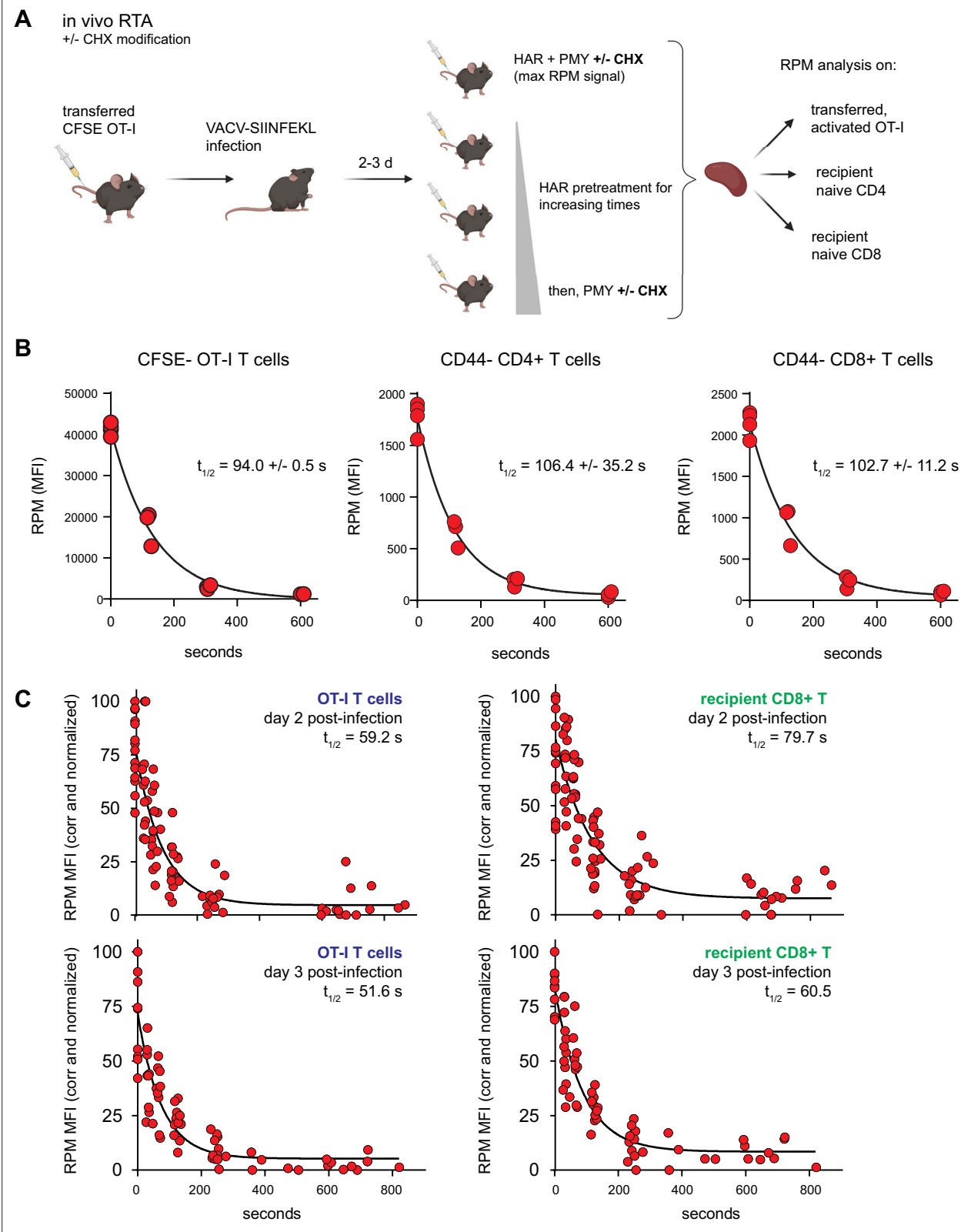

**Figure 4.** Translation rates of resting and activated T cells in vivo. (**A**) Depiction of the in vivo ribopuromycylation method (RPM)-ribosome transit assay (RTA) method. Labeled OT-I T cells are first adoptively transferred, followed by VACV-SIINFEKL infection of mice. RTA analysis is performed by intravenous injection of harringtonine (HAR) followed by puromycin (PMY; +/− cycloheximide [CHX] to prevent leakiness from HAR inhibition alone). Spleens are harvested for RPM analysis on both endogenous and transferred T cells. Schematic designed with Biorender. (**B**) Carboxyfluorescein

*Figure 4 continued on next page*

*Figure 4 continued*

succinimidyl ester (CFSE)-labeled Ly5.2+ (CD45.2+CD45.1−) OT-I T cells were adoptively transferred into Ly5.1 (CD45.1+CD45.2−) mice, which were then infected with VACV-SIINFEKL to activate the OT-I cells. Three days after infection, mice were intravenously injected with HAR simultaneously with PMY for 5 min (maximum signal), or first injected with HAR for ~110, ~275, or ~575 s before being injected with PMY for 5 min. Splenocytes from mice were harvested, surface stained for gating and activation markers as indicated, fixed and permeabilized, and stained for RPM. Gates were CFSE^low OT-I CD8+ T cells to measure decay in activated cells, and CD44−CD8+ or CD44−CD4+ T cells to measure decay in resting T cells. The curve was generated by fitting to a one phase exponential decay. Representative of two independent experiments, 2–4 mice per group, with the mean and standard deviation of the calculated half-life decays as indicated. (**C**) RTA, with the CHX modification, of adoptively transferred OT-I T cells or un-activated host CD8+ T cells in mice infected for 2 or 3 days with VACV-SIINFEKL. Three to four independent experiments combined, normalized by setting maximum background-subtracted signal to 100.

The online version of this article includes the following source data and figure supplement(s) for figure 4:

**Source data 1.** Numerical data and statistics related to *Figure 4*.

**Figure supplement 1.** Ribopuromycylation method (RPM) cell phenotyping and in vivo T cell division.

**Figure supplement 1—source data 1.** Numerical data and statistics related to *Figure 4—figure supplement 1*.

the methodological adaptation enhances activation. Direct comparison of the protocols confirmed the superiority of Tan et al ., as shown by activation markers (CD69, CD25, CD44), cell size (measured by side scatter, which correlates well with automated diameter measurements), and cell division (*Figure 5—figure supplement 1A,B*). OT-I cells activated by this protocol yielded large increases in the observed polysome fractions of activated splenocytes, or lymph-node-derived lymphocytes (*Figure 5B*). This was also evident during high salt fractionation conditions, where we found that 500 mM was necessary to fully dissociate non-translating ribosomes compared to the often used 300 mM (*Figure 5—figure supplement 2A,B*). T cell ribosomes had quantifiable but low levels of monosomes under these high salt conditions (*Figure 5—figure supplement 2C,D*; note that we could not obtain enough activated OT-I cells in vivo for these experiments).

These findings indicate that monosomes make a major contribution to translation in resting T cells but are likely to make a minor contribution in fully activated cells. These results might also complicate the conclusion reached by *Gerashchenko et al., 2021* that HAR may only be useful until 45 s after the start of treatment, as the assumption was made that polysomes were the only ribosome subset actively translating mRNA.

## Accounting for translation in lymphocytes: measuring the protein-to-ribosome ratio

Cells need to synthesize sufficient proteins to regenerate a complete proteome each division cycle. This number will depend on the division rate, cell size, protein concentration, and protein loss due to degradation and export (secretion, release of exosomes, loss of other cellular material). To understand how the protein synthesis apparatus enables such rapid T cell division times, we quantitated a number of critical protein synthesis parameters in resting and activated OT-I T cells (*Figure 6A*). For these experiments, we used the optimized protocol for in vitro OT-I T cell activation (*Tan et al., 2017*) that greatly increased the fraction of ribosomes in polysomes on day 1 post activation.

Automated microscope measurements revealed that OT-I T cells increase in diameter from the resting state to the day 1 and 2 activated states, with a corresponding calculated volume increase (based on spherical geometry) of ~2.9-fold (*Figure 6B*). To quantify protein content, we determined total tryptophan (Trp) autofluorescence of fully denatured proteins in a total cell lysate (*Wiśniewski and Gaugaz, 2015*). Protein content per T cell increases ~fivefold following activation (*Figure 6C*), from 421 million proteins per cell (assuming an average length of 472 aa and a proteome Trp content of 0.69% *Wiśniewski and Gaugaz, 2015*) to 2.15 billion proteins per cell in day 2 activated cells, resulting in a net 1.7-fold increase in protein concentration (*Figure 6D*).

We determined the number of ribosomes per cell using a Bioanalyzer electrophoresis device to measure the amount of 18S and 28S rRNA in purified total RNA based on staining with a RNA-binding dye and utilizing a spike-in standard mRNA to control for yield loss during RNA purification (*Figure 6E*). The maximal number of translating ribosomes is limited by the less abundant subunit,

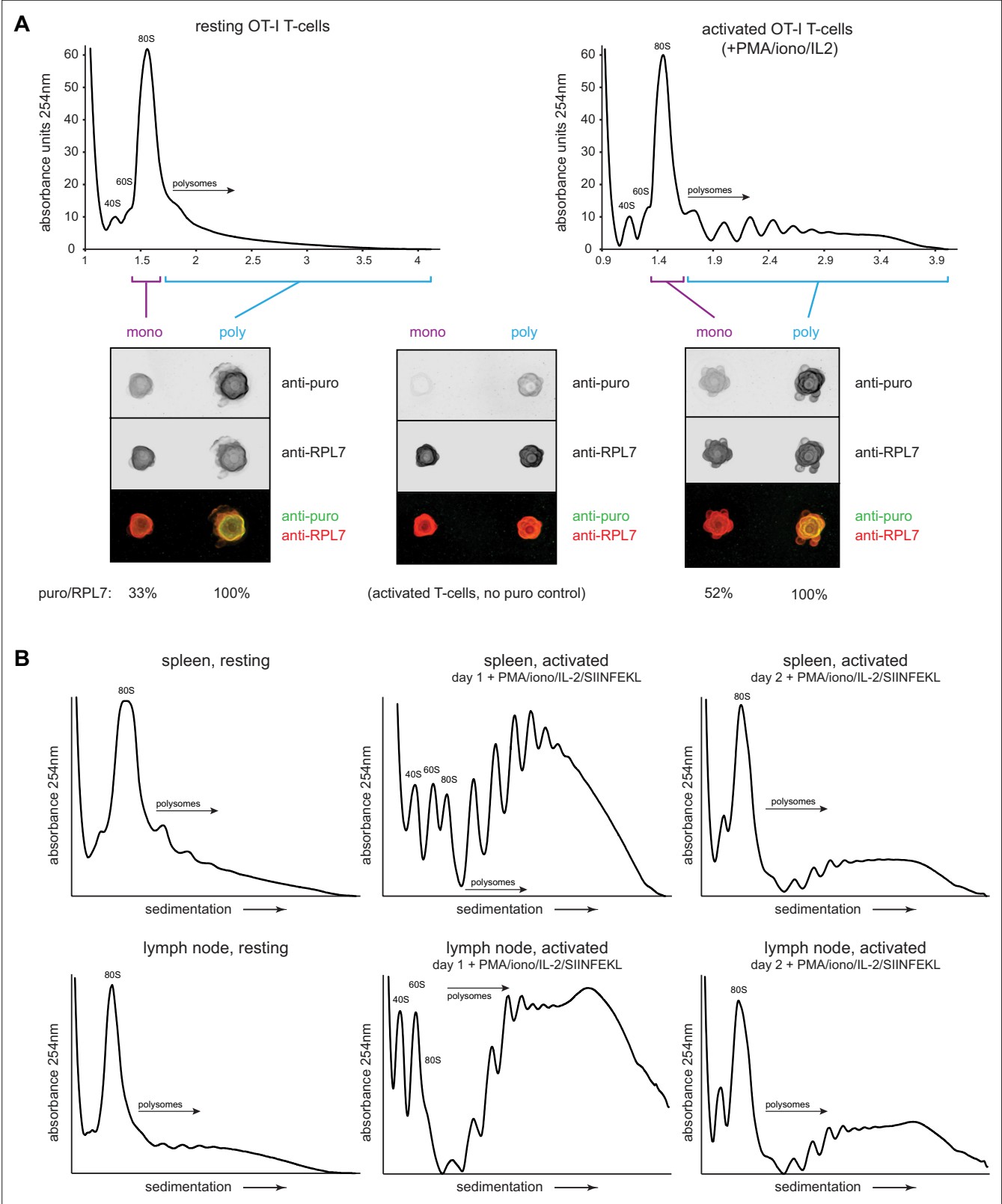

**Figure 5.** Puromycylation reveals percentage of actively translating monosomes in resting and activated T cells. (**A**) OT-I mice were treated intravenously with cycloheximide (CHX) and puromycin (PMY), and lymphocytes from the spleens and lymph nodes were isolated and subjected to polysome profiling by ultracentrifugation through 15–45% sucrose gradients (resting OT-I T cells). OT-I T cells activated in vitro for 2 days with PMA/ionomycin and IL-2 (without cognate SIINFEKL peptide) were treated either with CHX alone (no PMY control) or CHX with PMY and subjected to polysome profiling. The

*Figure 5 continued on next page*

*Figure 5 continued*

indicated fractions were collected, pooled, and their ribosomes were re-isolated and dotted onto a nitrocellulose membrane for blotting with antibodies against PMY and RPL7. After subtraction of background signal from the anti-puro antibody (middle panel), the PMY/RPL7 ratio of monosomes was expressed relative to that of polysomes, which was defined as 100% translating. Representative of two biological replicates. (**B**) For resting T cells, OT-I mice were treated intravenously with CHX, and lymphocytes from the spleens or lymph nodes were isolated and lysed. For activated T cells, lymph node or splenic OT-I T cells were stimulated in vitro for 2 days with PMA/ionomycin, IL-2, and exogenous SIINFEKL, followed by treatment with CHX for 5 min. For both resting and activated cells, ribosome-containing lysates were fractionated via ultracentrifugation on 15–45% sucrose gradients. Representative of two biological replicates.

The online version of this article includes the following source data and figure supplement(s) for figure 5:

**Source data 1.** Numerical data related to *Figure 5*.

**Source data 2.** Uncropped and outlined immunoblot images related to *Figure 5A*.

**Figure supplement 1.** Exogenous SIINFEKL significantly enhances OT-I T cell activation in vitro.

**Figure supplement 1—source data 1.** Numerical data related to *Figure 5—figure supplement 1*.

**Figure supplement 2.** Salt stability of T cell ribosomes and monosome quantification.

**Figure supplement 2—source data 1.** Numerical data related to *Figure 5—figure supplement 2*.

---

which in all cases is the 60S subunit (typically 75–90% of the 40S subunit). 60S subunits increased both in absolute terms and per unit cell volume as T cells became activated, reaching a maximum of ~3.6 million copies per T cell following 2d in vitro activation from 264,000 copies in resting T cells (*Figure 6F–G*). These numbers are similar to those reported by *Wolf et al., 2020*, but should be more accurate since Wolf et al. used total cellular RNA content to estimate ribosomes.

Could there be a significant pool of non-functional ribosomes in the nucleus, where initial assembly occurs, and which occupies nearly 50% of the volume of resting T lymphocytes (*Petrzilka et al., 1978*) and 34% of activated T cells (*Petrzilka and Schroeder, 1979*)? Immunoblotting of fractionated nuclei shows that the distribution of ribosomes in lymphocytes is similar in resting and activated OT-I T and HeLa cells, with only a small fraction of total ribosomal subunits detected in nuclear lysates (*Figure 6—figure supplement 1*).

The ratio of proteins to ribosomes is critical since it dictates the minimal time to replicate the proteome during cell division. This dropped up to threefold as T cells became activated (*Figure 6H*). Since mammalian ribosomes elongate at ~6 residues per second (*Ingolia et al., 2011*; *Fan and Penman, 1970*), we calculated the minimal time for a ribosome to recreate the proteome based on the protein/maximally assembled ribosome ratios, not accounting for protein degradation or secretion.

For HeLa cells, it would take a ribosome 19.9 hr to synthesize 910 'average' proteins of 472 amino acids, reasonably close to the reported doubling time of ~24 hr. For OT-I T cells, with an in vitro doubling time of ~9.7 hr, the calculated minimal proteome duplication time is also within shouting distance – 10.1 hr by day 2. Therefore, the division rates of in vitro activated OT-I T cells, and HeLa cells can be approximated from the number of proteins and functional ribosomes translating a full capacity.

## Paradoxical discrepancy in OT-I cell division rate and protein synthesis capacity

We extended these findings to OT-I T cells in vivo, determining first that adoptively transferred OT-I T cells divide most rapidly between days 1 and 2 of activation during acute viral infection, with an average doubling time of 6.8 hr, slowing to approximately 7.7 hr by day 2 post-infection (via CFSE labeling; *Figure 4—figure supplement 1C–E*). We sorted for transferred OT-I T cells on day 2 post-infection and measured cell size, protein, and ribosome numbers ('ex vivo day 2' measurements in *Figure 6* graphs). Cells activated in vivo were similar in size and protein content to in vitro activated cells, but the protein-to-ribosome ratio was significantly higher than in vitro activated T cells due to the presence of 2.3 vs. 3.6 million 60S subunits in maximally in vitro activated T cells.

Remarkably, the ratio of proteins to ribosomes (1017) at this juncture dictates a minimal proteome duplication time of 22.2 hr, nearly 3× the measured doubling time of 7.7 hr. While our RTA measurements support a higher elongation rate in vivo ($t_{1/2}$ = 55 vs. 70 s in HeLa cells), the 27% increase (7.6 residues per second) does not come close to accounting for the discrepancy. Thus, a paradox: protein synthesis activity or capacity of in vivo activated T cells does not support their doubling times.

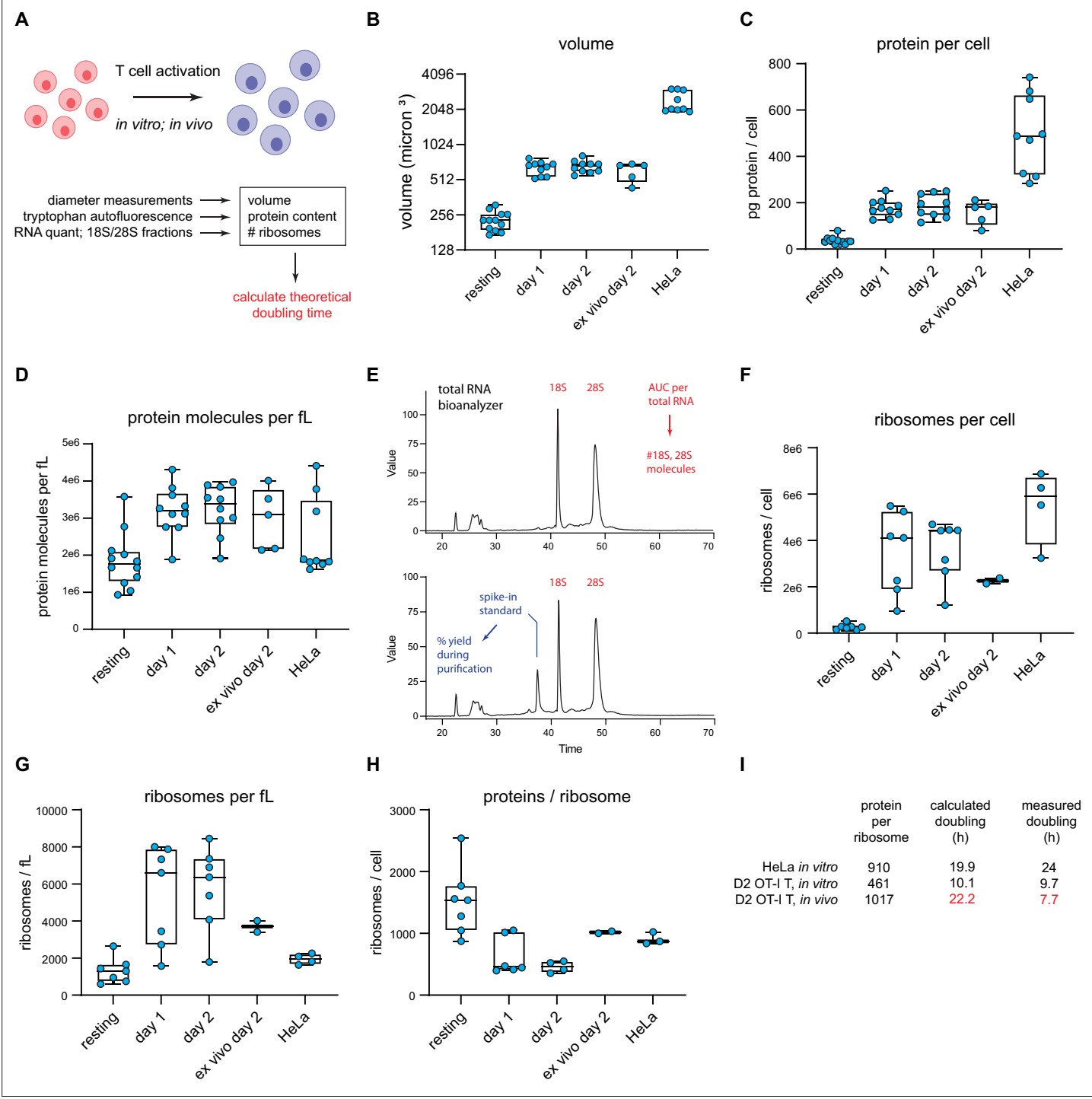

**Figure 6.** T cell accounting reveals discrepancy in proteome duplication rate for activated T cells in vivo. (**A**) Measurements made to calculate in vitro and in vivo rates of T cell division. (**B**) Volume calculations based on diameter measurements made by automated cell counter for the indicated cell types. Days 1 and 2 represent in vitro activated OT-I T cells. Ex vivo day 2 represent cells activated in vivo for 2 days, followed by isolation and processing. (**C**) Protein content per cell as measured by tryptophan fluorescence of denatured lysates. (**D**) Protein molecules per fL, assuming an average protein length of 472 aa and average amino acid mass of 110 Da. (**E**) Example output from custom bioanalyzer method to determine number or ribosomes per cell. Total RNA is quantified and the bioanalyzer is used to determine area under the curve for 18S and 28S percentage of total RNA. Additionally, an exogenous mRNA standard is spiked into the sample prior to RNA isolation to determine the percent loss in yield during the purification procedure. Combined, this method allows for the accurate determination of total number of 18S and 28S molecules per cell. (**F**) Number of ribosomes per cell for the indicated cells. (**G**) Ribosome per fL for the indicated cells. (**H**) The protein/ribosome ratio, a representation of how many

*Figure 6 continued on next page*

*Figure 6 continued*

proteins a single ribosome would need to create to duplicate the proteome. (**I**) Discrepancy between measured and calculated rates of division for OT-I T cells activated and dividing in vivo.

The online version of this article includes the following source data and figure supplement(s) for figure 6:

**Source data 1.** Numerical data related to *Figure 6*.

**Figure supplement 1.** Fractionation of HeLa or T cells reveals few ribosomal components in nuclear lysates.

**Figure supplement 1—source data 1.** Uncropped and outlined immunoblot images related to *Figure 6—figure supplement 1*.

**Figure supplement 1—source data 2.** Uncropped immunoblot images related to *Figure 6—figure supplement 1*.

## Discussion

Lymphocytes protect jawed vertebrates against viral and cellular microbes and tumors. To counter the essentially infinite diversity of antigens expressed by pathogens, lymphocytes evolved to generate an enormous repertoire of specific antigen receptors. Hosts hope to never need the repertoire, and until a cognate antigen appears, metabolic processes, including protein synthesis, are minimized (this might also extend the life span of naive cells since the generation of defective ribosomal products (DRiPs) and other damaging chemical byproducts will also be minimized). Upon activation, lymphocytes divide rapidly to achieve numbers capable of exerting effective immunity. Here, we studied aspects of protein synthesis in lymphocytes, a field fairly dormant since the pioneering studies by the Kay and Cooper laboratories in the 70s but now experiencing a renaissance (*Tan et al., 2017*; *Wolf et al., 2020*; *Araki et al., 2017*; *Marchingo and Cantrell, 2022*; *Howden et al., 2019*).

To measure translation elongation rates in vivo and in vitro, we developed flow RTA, which is far simpler and cheaper than original (*Fan and Penman, 1970*) and recent (*Ingolia et al., 2011*; *Conn and Qian, 2013*) methods and provides information at the level of individual cells simultaneously phenotyped by standard flow cytometry markers. While this work was in progress, the Pierre lab described 'SunRiSE', a nearly identical approach, to measure elongation rates in vitro, observing similar puromycylation decay rates following HAR treatment (*Argüello et al., 2018*). While our findings are similar regarding the elongation rates of fibroblasts and lymphocytes, the addition of elongation inhibitors to the protocol (CHX or EME) greatly reduces the leakiness of HAR, thereby improving the calculated elongation rate accuracy.

RTA revealed exciting facets of lymphocyte translation. We find that a significant fraction of ex vivo lymphocytes possess stalled/primed ribosomes that puromycylate nascent chains but do not transit mRNA. *Graber et al., 2013* used the original RPM protocol to show that primary neurons possess substantial numbers of stalled ribosomes, apparently to facilitate rapid translation upon synaptic signaling. Otherwise, to our knowledge, such prolonged ribosomal stalling has not been described in mammalian cells. These experiments may also be useful in examining the phenomenon of 'poised mRNA', originally described in lymphocytes for cytokine mRNAs and more recently expanded on with advanced sequencing techniques (*Turner, 2023*; *Choi et al., 2022*).

Our improved RTA protocol reveals the dramatic upregulation of protein synthesis by OT-I CD8[+] T cells activated in vivo, with a 15-fold increase in translation in day 1 activated OT-I T cells vs. resting OT-I T cells. Activated T cells divide every 6.8 hr from day 1 to 2 post-VACV-SIINFEKL infection. This is consistent with a prior OT-I study in mice infected with a different SIINFEKL-expressing VACV (*Xiao et al., 2007*). Importantly, by i.v. delivery of translation inhibitors, we show that RTA can be used to measure elongation rates in vivo. Though we focus on lymphocytes, in vivo RTA can be used to study any cell type in animals that can be analyzed ex vivo by flow cytometry.

Contrary to observations in vitro, ribosomes are not stalled in naive mouse T cells in vivo, as we show via RTA analysis of non-activated T cells. Importantly, ribosome transit times were up to ~30% faster in activated cells, consistent with the idea that lymphocytes can accelerate translation to support activation and the rapid cell division that ensues. Similarly, OT-I T cells increased elongation rates in vitro when incubated at fever temperature (39.5°C). While such accelerated translation may decrease translational fidelity, the impact may be lessened by the terminal nature of lymphocyte division, since the vast majority of activated cells apoptose within weeks of activation.

We additionally provide initial measurements of numbers of ribosomes and their protein synthesis activity, key values in accounting for the macroeconomics of T cell protein synthesis. Of particular importance is the ratio of cellular proteins to ribosomes; in conjunction with the elongation rate, this

value dictates the minimal time (i.e., no protein degradation or secretion) for duplicating the proteome (*Shore and Albert, 2022*). Using the mammalian cell "speed limit" of 6 residues per second (*Ingolia et al., 2011*; *Fan and Penman, 1970*) mouse T cells do not appear to possess sufficient ribosomes to support a 6- to 8-hr division time. Even if ribosomes in in vivo activated T cells are translating at 7.8 residues per second, the time required to synthesize the proteome is 2.3× greater than the observed replication time (15.5 vs. 6.8 hr).

The discrepancy is further exacerbated when accounting for protein secretion and degradation of DRiPs (30% of nascent proteins) (*Qian et al., 2005*; *Princiotta et al., 2003*; *Schubert et al., 2000*; *Wheatley et al., 1980*) and retirees $t_{1/2}$ of ~32 hr over the entire proteome (*Doherty et al., 2009*; *Cambridge et al., 2011*) and the presence of stalled and resting ribosomes. Together, this likely doubles the time required to synthesize the proteome.

Something is obviously wrong. T cell doubling times of 6.8 hr are very likely to be accurate, as they are simple to measure and are routinely reported in mouse T cell studies. Quantitating proteins, however, is more challenging than it might seem. Where are the potential gremlins?

1. *Quantitating cellular protein*. We initially used the various dye-binding assays for quantitating cellular protein content. While these assays vary notoriously for quantitating different proteins, they provide similar values for the cellular proteome. Moreover, they were in good agreement with a completely independent method based on Trp fluorescence, which we consider the gold standard for protein quantitation (*Wiśniewski and Gaugaz, 2015*). We use Wisniewski and Gaugaz's value for Trp abundance in the proteome (0.69%) and note that this value is nearly identical to values obtained using the abundance of Trp toted up from proteomic analysis of 11 different human tumor cell lines (*Geiger et al., 2012*) as well as a transgenic T cell (*Hukelmann et al., 2016*). One possible source of error is the free metabolic pool of tryptophan, but this is likely to be less than 5% of protein tryptophan (*Kane et al., 1999*). Another is the presence of serum proteins in cellular lysates. These are unlikely, however, to significantly contribute to our values since we fail to see fetal bovine serum-derived bovine serum albumin (BSA) in sodium dodecyl sulfate–polyacrylamide gel electrophoresis (SDS–PAGE) of total cell lysates.

2. *Quantitating ribosomes*. We originally quantitated ribosomes using antibodies specific for ribosomal proteins in immunoblots of total cell lysates with purified ribosomes as a standard. We eventually recognized, however, that this approach is limited by heterogeneity in ribosome composition (*Genuth and Barna, 2018*; *Slavov et al., 2015*) , as well as the presence of free pools of any given ribosomal subunit. It dawned on us that ribosomes are, in principle, simple to quantitate based on their RNA species, which account for >80% of total cellular RNA. Though we initially quantitated rRNA species on agarose gels with purified ribosomes as a standard, we believe quantitation is more accurate using a Bioanalyzer with doped-in highly purified RNA as an absolute staining standard and a control for yield loss during purification of samples.

3. *Ribosome elongation rates*. The classical value of ~6 or fewer residues per second for radiolabeling studies of cultured cells seems likely to be accurate based on ribosome profiling (*Ingolia et al., 2011*; *Ingolia, 2014*). A recent ribosome profiling study extends these findings to mice, with in vivo elongation rates of 6.8, 5.2, and 4.4 amino acids per second for liver, kidney, and skeletal muscle, respectively (*Gerashchenko et al., 2021*). Our in vivo RTA data demonstrate that translation elongation in in vivo activated T cells is 30% faster than in cultured cells and thus is likely to be up to ~7.8 residues/s. We further note that it is likely that puromycylation detects only a subset of nascent chains. Indeed, in dozens of studies (including our *Figure 2C*), immunoblots of puromycylated proteins detect discreet bands in SDS–PAGE gel rather than the expected smear if all chains are randomly puromycylated at all lengths. This may be due to non-random incorporation of PMY, non-random antibody detection of puromycylated nascent chains, or a combination of both. Though it seems improbable, it is possible that this bias influences RTA inferred elongation rates.

While one or more of these values may yet be inaccurate, we note that *Wolf et al., 2020* mass spectrometric-based measurements of in vitro T cell protein synthetic capacity supports and even exacerbates the paradox. We must therefore consider the possibility that that lymphocytes are in such a hurry to divide that they resort to the extraordinary measure of acquiring proteins from resting lymphocytes or other cell types.

There are reports that neurons acquire ribosomes from Schwann cells (*Court et al., 2008*; *Court et al., 2011*) and that cancer cells acquire mitochondria from immune cells (*Saha et al., 2022*). Furthermore, through trogocytosis, lymphocytes acquire cell surface molecules from other cells (*Joly and*

*Hudrisier, 2003*). We are, however, proposing that lymphocytes acquire a significant fraction of their proteome, perhaps via something akin to emperipolesis or entosis (*Yang and Li, 2012*), where cells actively enter homotypic cells and can even divide while residing inside (*Overholtzer et al., 2007*).

In any event, our findings clearly indicate how much remains to be learned about basic lymphocyte cell biology and the importance of simple accounting in squaring our models of cell biology with reality.

# Materials and methods

## Key resources table

| Reagent type (species) or resource | Designation | Source or reference | Identifiers | Additional information |
|---|---|---|---|---|
| Strain, strain background (*Mus musculus,* male, female) | (*C57BL/6J*) male, female | The Jackson Laboratory | RRID:IMSR_JAX:000664 Strain # (000664) | (6 weeks to 8 months of age) |
| Strain, strain background (*Mus musculus,* male, female) | (*C57BL/6NTac*) male, female | Taconic Biosciences | RRID:IMSR_TAC:B6 Model # (B6-M, B6-F) | (6 weeks to 8 months of age) |
| Strain, strain background (*Mus musculus,* male, female) | (*B6.SJL-Ptprc$^a$/BoyAiTac*) male, female | Taconic Biosciences | RRID:IMSR_TAC:1349 Model # (4007-M, 4007-F) | (6 weeks to 8 months of age) CD45.1 (Ly5.1) |
| Strain, strain background (*Mus musculus,* male, female) | Strain (*B6.129S7-Rag1$^{tm1Mom}$ Tg(TcraTcrb)1100Mjb N9 + N1*) male, female | NIAID Intramural Research Repository | RRID:IMSR_TAC:4175 Model # (4175-M, 4175-F) | (6 weeks to 8 months of age) RAG1ko OTI TCR transgenic |
| Cell line (*Homo sapiens*) | HeLa, Epithelial cell, uterus; cervix (adenocarcinoma) | ATCC | RRID:CVCL_0030 CCL2 | |
| Cell line (*Homo sapiens*) | Peripheral blood mononuclear cells (PBMCs, healthy, adult) | NIH Clinical Center Department of Transfusion Medicine | | |
| Chemical compound, drug | Puromycin | Calbiochem | 540222 – 100 MG | (1 mg/ mouse) (20 µg/ml) |
| Chemical compound, drug | Cycloheximide | EMD Millipore | 239764 – 100 MG | (0.34 mg/mouse) (200 µg/ml) |
| Chemical compound, drug | Harringtonin | Santa Cruz Biotech | sc-204771A | (100 µg/mouse) (5 µg/ml) |
| Chemical compound, drug | Emetine, dihydrochloride | Calbiochem | 324693 | (25 µg/ml) |
| Chemical compound, drug | Anisomycin | Sigma | A9789-25MG | (50 µg/ml) |
| Chemical compound, drug | Pactamycin | Sigma | PZ0182 | (10 µM) |
| Chemical compound, drug | Carboxyfluorescein succinimidyl ester (CFSE) | Invitrogen | C34554 | (5 µM) |
| Antibody | Anti-human CD3 mouse monoclonal antibody (OKT3), PerCP-eFluor 710, eBioscience | Invitrogen | Cat#: 46-0037-42 RRID:AB_1834395 | FACS (1 µl per test) |
| Antibody | Anti-human mouse CD19 Monoclonal Antibody (HIB19), PE, eBioscience | Invitrogen | Cat#: 12-0199-42 RRID:AB_1834376 | FACS (1 µl per test) |
| Antibody | Anti-human mouse CD45 Monoclonal Antibody (2D1), APC-eFluor 780, eBioscience | Invitrogen | Cat#: 12-0199-42 RRID:AB_1834376 | FACS (1 µl per test) |
| Antibody | BD Horizon PE-CF594 Mouse anti-Human CD4 (L3T4) | BD Biosciences | Cat#: 562281 RRID:AB_11154597 | FACS (1 µl per test) |
| Antibody | Anti-human mouse CD4 Monoclonal Antibody (RPA-T4), PE-Cyanine7, eBioscience | Invitrogen | Cat#: 25-0049-42 RRID:AB_1659695 | FACS (1 µl per test) |

*Continued on next page*

*Continued*

| Reagent type (species) or resource | Designation | Source or reference | Identifiers | Additional information |
|---|---|---|---|---|
| Antibody | BD Horizon BV421 Mouse Anti-Human CD8 (RPA-T8) | BD Biosciences | Cat#: 562428 RRID:AB_11154035 | FACS (1 µl per test) |
| Antibody | BD Horizon BV786 Hamster Anti-Mouse CD3e (145 – 2 C11) | BD Biosciences | Cat#: 564379 RRID:AB_2738780 | FACS (0.75 µl per test) |
| Antibody | BD Horizon BV510 Rat Anti-Mouse CD4 (RM4-5) | BD Biosciences | Cat#: 563106 RRID:AB_2687550 | FACS (0.75 µl per test) |
| Antibody | BD Horizon APC-R700 Rat anti-Mouse CD5 (53 – 7.3) | BD Biosciences | Cat#: 565505 | FACS (0.75 µl per test) |
| Antibody | BD Horizon PE-CF594 Rat Anti-Mouse CD8a (53 – 6.7) | BD Biosciences | Cat#: 562283 RRID:AB_11152075 | FACS (0.75 µl per test) |
| Antibody | Anti-mouse rat CD11b Monoclonal Antibody (M1/70), PE-Cyanine7, eBioscience | Invitrogen | Cat#: 25-0112-82 RRID:AB_469588 | FACS (0.75 µl per test) |
| Antibody | Anti-mouse rat CD19 Monoclonal Antibody (eBio1D3 (1D3)), APC, eBioscience | Invitrogen | Cat#: 17-0193-82 RRID:AB_1659676 | FACS (0.75 µl per test) |
| Antibody | BD Horizon BV650 Rat Anti-Mouse CD25 (PC61) | BD Biosciences | Cat#: 564021 RRID:AB_2738547 | FACS (0.75 µl per test) |
| Antibody | BD Horizon BV605 Rat Anti-Mouse CD44 (IM7) | BD Biosciences | Cat#: 563058 RRID:AB_2737979 | FACS (0.75 µl per test) |
| Antibody | Anti-mouse rat CD44 Monoclonal Antibody (IM7), eFluor 450, eBioscience | Invitrogen | Cat#: 48-0441-82 RRID:AB_1272246 | FACS (0.75 µl per test) |
| Antibody | Anti-mouse mouse CD45.1 Monoclonal Antibody (A20), APC, eBioscience | Invitrogen | Cat#: 17-0453-82 RRID:AB_469398 | FACS (0.75 µl per test) |
| Antibody | Anti-mouse mouse CD45.1 Monoclonal Antibody (A20), eFluor 450, eBioscience | Invitrogen | Cat#: 48-0453-82 RRID:AB_1272189 | FACS (0.75 µl per test) |
| Antibody | Anti-mouse Armenian hamster TCR gamma/delta Monoclonal Antibody (eBioGL3 (GL-3, GL3)), PE, eBioscience | Invitrogen | Cat#: 12-5711-82 RRID:AB_465934 | FACS (0.75 µl per test) |
| Antibody | BD Pharmingen PE Rat Anti-Mouse Ly-6G and Ly-6C (RB6-8C5) | BD Biosciences | Cat#: 561084 RRID:AB_394644 | FACS (0.75 µl per test) |
| Antibody | Anti-mouse mouse NK1.1 Monoclonal Antibody (PK136), FITC, eBioscience | Invitrogen | Cat#: 11-5941-82 RRID:AB_465318 | FACS (0.75 µl per test) |
| Antibody | BD Horizon BV711 Hamster Anti-Mouse TCR β Chain (H57-597) | BD Biosciences | Cat#: 563135 RRID:AB_2738023 | FACS (0.75 µl per test) |
| Antibody | BD Pharmingen PE Mouse Anti-Mouse Vβ 5.1, 5.2 T-Cell Receptor (MR9-4) | BD Biosciences | Cat#: 562086 RRID:AB_394698 | FACS (0.75 µl per test) |
| Antibody | Anti-puromycin (PMY-2A4) | (made in-house) Developmental Studies Hybridoma Bank | Cat#: PMY-2A4 RRID:AB_2619605 | FACS (1 µl per test) |
| Antibody | Human anti-riboP | Immunovision | PAG-3000 | Immunoblotting |
| Antibody | Rabbit anti-RPL7 | Abcam | Catalog number: ab72550, RRID:AB_1270391 | Immunoblotting |

*Continued on next page*

*Continued*

| Reagent type (species) or resource | Designation | Source or reference | Identifiers | Additional information |
|---|---|---|---|---|
| Antibody | Rabbit anti-RPL26 | Bethyl Laboratories | Catalog number: A300-686A, RRID:AB_530289 | Immunoblotting |
| Antibody | Mouse anti-beta actin | Invitrogen | Catalog number: MA1-140, RRID:AB_2536844 | Immunoblotting |
| Antibody | Rabbit anti-HSP90 | Santa Cruz | Discontinued | Immunoblotting |
| Antibody | Rat anti-GRP94 | Enzo | Catalog number: ADI-SPA-850-D, RRID:AB_2039133 | Immunoblotting |
| Antibody | Rabbit anti-RPL28 | Abcam | Catalog number: ab138125 | Immunoblotting |
| Antibody | Rabbit anti-RPL6 | Abcam | Catalog number: ab176705 | Immunoblotting |
| Antibody | Mouse anti-PDI | Abcam | Catalog number: ab2792, RRID: AB_303304 | Immunoblotting |
| Antibody | Mouse anti-lamin A/C | Cell Signaling Technology | Catalog number: 4777, RRID:AB_1054575 | Immunoblotting |
| Antibody | Rabbit anti-fibrillarin | Cell Signaling Technology | Catalog number: 2639, RRID: AB_2278087 | Immunoblotting |
| Antibody | Mouse anti-RPS6 | Cell Signaling Technology | Catalog number: 2317, RRID:AB_2238583 | Immunoblotting |
| Antibody | Rabbit anti-histone H3 | Cell Signaling Technology | Catalog number: 4499, RRID:AB_10544537 | Immunoblotting |
| Antibody | Rabbit anti-RPL5 | Cell Signaling Technology | Catalog number: 51345, RRID:AB_279939 | Immunoblotting |
| Antibody | Rabbit anti-RPS3 | Cell Signaling Technology | Catalog number: 9538, RRID:AB_10622028 | Immunoblotting |
| Commercial assay or kit | Alexa Flour 647 Protein Labelling Kit | Invitrogen | Cat#: A20173 | Used to label anti-puromycin Ab, used at 1 µl per test for FACS |
| Peptide, recombinant protein | SIINFEKL | Research Technology Branch, NIAID | N/A | For T cell activation |
| Strain, strain background | vaccinia virus NP (366NT60)-SIINFEKL-tdTomato | This paper | N/A | |
| Chemical compound/drug | Leucine, L-[4,5-$^3$H], 1 mCi | Revvity | Cat.#: NET1166001MC | (0.2 mCi/ml) |
| Commercial assay or kit | *DC* Protein Assay Kit I | Bio-Rad | Cat.#: 5000111 | Protein quantification |
| Recombinant DNA reagent | CleanCap EGFP mRNA | Tri-Link | Cat.#: L-7601 | Spike in standard for RNA quantification |
| Commercial assay or kit | Agilent RNA 6000 Nano Kit | Agilent | Cat.#: 5067 – 1511 | RNA quantification |

## Mice

Specific pathogen-free C57BL/6 mice were purchased from the Jackson Laboratory or from Taconic. OT-I TCR transgenic mice were acquired from the NIAID Intramural Research Repository. All mice were housed under specific pathogen-free conditions (including murine norovirus, mouse parvovirus, and mouse hepatitis virus) and maintained on standard rodent chow and water supplied ad libitum. All animal studies were approved by and performed in accordance with the Animal Care and Use Committee of the National Institute of Allergy and Infectious Diseases under protocol LVD-5E. For acute infections, and to generate memory T cells, CFSE-labeled Ly5.2+ (CD45.2$^+$CD45.1$^-$) OT-I T cells were adoptively transferred into Ly5.1 (CD45.1$^+$CD45.2$^-$) mice. A subset of these mice was infected with VACV-SIINFEKL for indicated times to activate OT-I T cells, with some mice left uninfected where

specified. For experiments done directly on memory OT-I T cells, assays were done 8–9 weeks after infection.

## Cells

HeLa cells were obtained originally from the ATCC, were authenticated by the ATCC STR profiling service, and were routinely confirmed to be mycoplasma-negative with the ATCC Universal Mycoplasma Detection Kit.

## In vivo RPM, in vivo RPM-RTA, and relative protein synthesis determination

For the standard and CHX-improved in vivo RPM assays, mice were intravenously injected with 100 µl of a 10 mg/ml solution of PMY in phosphate-buffered saline (PBS) that was warmed to 37°C, or PMY, as just described, along with 0.34 mg per mouse of CHX. After indicated times, mice were sacrificed, and organs were collected into complete Roswell Park Memorial Institute (RPMI) on ice (Gibco RPMI supplemented with 7.5% fetal calf serum). For the in vivo RTA, or the CHX-improved in vivo RTA, mice were intravenously injected with 100 µg of HAR simultaneously with 1 mg of PMY, or 1 mg of PMY and 0.34 mg of CHX for 5 min (for the maximum signal) or first intravenously injected with 100 µg of HAR for the times indicated before being intravenously injected with 1 mg of PMY, or PMY and 0.34 mg of CHX for 5 min. To determine relative levels of active protein synthesis, two sets of mice were required. In the first set, mice were intravenously injected with 100 µg of HAR for 15 min, and then intravenously injected with 1 mg of PMY and 0.34 mg of CHX for 5 min before spleens were harvested. In the second set, mice were intravenously injected simultaneously with 0.34 mg of CHX and 1 mg of PMY for 5 min before spleens were harvested.

## Single-cell preparation from organs

Isolated organs were crushed between two frosted microscope slides, and the resultant single cell suspension was filtered through a 70-µm mesh screen. The filtered single cell suspension was then centrifuged, resuspended in ACK lysing buffer (Lonza) to lyse red blood cells, centrifuged again, and resuspended in complete RPMI for counting on a Nexcelom Cellometer Vision using Trypan Blue (Lonza BioWhittaker) for live/dead cell discrimination and cell diameter measurements.

## CFSE labeling

Spleens and inguinal, mediastinal, cervical, mesenteric, and popliteal lymph nodes from OT-I TCR transgenic or C57BL/6 mice were processed into a single-cell suspension, red blood cells were lysed in ACK lysing buffer, and the resultant cells filtered through a 70-µM mesh screen. After two washes in PBS, cells were counted on a Nexcelom Cellometer Vision using Trypan Blue for dead cell exclusion, and cells were labeled in 5 µM CFSE (Invitrogen) in PBS at $1 \times 10^7$ cells per ml for 18 min in a 37° water bath with mixing every 6 min. Cells were washed three times in PBS, recounted, and adoptively transferred into the indicated mice or cultured as specified.

## Human lymphocyte purification and culture conditions for human and mouse lymphocytes

Elutriated human lymphocytes were from healthy anonymous donors at the NIH Clinical Center Department of Transfusion Medicine. After collection, elutriated lymphocytes were purified on a discontinuous 35–70% Percoll (Amersham Biosciences) gradient and washed once with ACK lysing buffer (Life Technologies) to remove contaminating red blood cells. For time-course experiments, purified lymphocytes were resuspended in PBS and labeled with CFSE where indicated (as described above) to enable tracking of cell division over time. Lymphocytes were plated at $1–2 \times 10^6$ cells/ml in RPMI: RPMI 1640 (Gibco) supplemented with 15% fetal calf serum (FCS), 25 mM 4-(2-hydroxyethyl)-1-piperazineethanesulfonic acid (HEPES, Corning Cellgro), 1 mM sodium pyruvate (Gibco), and 55 µM beta-mercaptoethanol(Gibco). Depending on the experiment, media was also supplemented with recombinant human IL-2 (BRB NCI Frederick, 25 U/ml), PMA (Sigma, 1 ng/ml), and ionomycin (Sigma, 100 ng/ml). For OT-I T cell cultures, PMA was added at 100 ng/ml instead, and, where noted, SIINFEKL was added as well (100 nM) for optimal activation. Lymphocytes were cultured in 6% $CO_2$ at 37°C and allowed to sit overnight prior to any experiments unless noted (noted as 'freshly isolated'). For

time-course experiments, lymphocytes were cultured for up to 5 days and resuspended in fresh media every 2 days. Cell counts, diameters, and viabilities (through Trypan blue exclusion) were made on a Nexcelom Cellometer Vision cell counter. Cell volumes were calculated assuming spherical geometry.

## In vitro RPM and RPM staining

For each sample, cells were resuspended at $2 \times 10^7$ cells per ml and 100 µl transferred into 96-well round-bottom plates. When indicated, the media contained protein synthesis inhibitors at the following concentrations: 5 µg/ml HAR (Santa Cruz Biotechnology), 25 µg/ml EME (Calbiochem), 200 µg/ml CHX (Sigma), 50 µg/ml anisomycin (Sigma), or 10 µM pactamycin (Sigma). After a 15-min incubation at 37°C, 50 µl of 3× PMY (Calbiochem) media was added (150 µg/ml, for a final concentration of 50 µg/ml) and the cells were incubated for an additional 5 min before shifting to ice and adding 100 µl of cold PBS. Cells were then stained with ethidium monoazide (10 µg/ml in PBS, Molecular Probes) for live/dead cell discrimination. After thorough washing, and a 10-min incubation with heat-inactivated sera, or 2.4G2 to block Fc receptors, cell surface antigens were labeled for 30 min at 4°C with the following antibodies: For human lymphocyte stains, antibodies against: CD3ε PerCP-eFluor 710 (clone OKT3, eBioscience), CD19 PE (clone HIB19, eBioscience), CD45 APC-eFluor 780 (clone 2D1, eBioscience), CD4 PE-CF594 (clone RPA-T4, BD) or CD4 PE-Cy7 (clone RPA-T4, eBioscience), and CD8α BV421 (clone RPA-T8, BD). For mouse lymphocyte stains, antibodies were: CD3ε BV786 (clone 145 – 2 C11, BD), CD4BV510 (clone RM4-5, BD), CD5 APC-R700 (clone 53-7.3, BD), CD8α PE-CF594 (clone 53-6.7, BD), CD11b PE-Cy7 (clone M1/70, eBioscience), CD19 APC-Cy7 (eBio1D3, eBioscience), CD25 BV650 (PC61, BD), CD44 BV605 (IM7, BD), CD44 eFl450 (clone IM7, eBioscience), CD45.1 APC (clone A20, eBioscience), CD45.1 eFl450 (clone A20, eBioscience), CD45.2 eFluor450 (clone 104, eBioscience), CD45.2 PE-Cy7 (clone 104, eBioscience), CD69 PerCP-Cy5.5 (clone H1.2F3, Invitrogen), γδ TCR PE (eBioGL3, GL3, eBioscience), Gr1 PE (clone RB6-8C5, BD), NK1.1 FITC (clone PK136, eBioscience), TCRβ 711 (H57-597, BD), and Vβ5.1/Vβ5.2 PE (clone MR9-4, BD). All antibodies were used at 1:150 dilution in buffered saline supplemented with 0.1% BSA. Next, cells were simultaneously fixed and permeabilized in fix/perm buffer (1% paraformaldehyde, 0.0075% digitonin in PBS) for 20 min at 4°C. Intracellular PMY was labeled with an anti-PMY antibody (clone 2A4) directly conjugated with Alex Fluor 647 (conjugated using the Life Technologies Protein Labeling Kit per the manufacturer's instructions) for 1 hr. Cells were thoroughly washed and resuspended in buffered saline supplemented with 0.1% BSA, flow cytometry performed on a BD LSRII or BD LSRFortessa X-20, and resulting data analyzed with FlowJo software. To gate on OT-I T CD8$^+$ T cells, setup was: singlets by FSCa and FSCw, lymphocytes by SSCa and FSCa, EMA$^-$ (live/dead cell marker), CD3$^+$CD19$^-$, CD8$^+$CD4$^-$, CD45.2$^+$CD45.1$^-$, and Vb5$^+$, and activation markers as indicated. For thymocyte subsets, gating setup was singlets by FSCa and FSCw, lymphocytes by SSCa and FSCa, EMA$^-$, and then subsets on combinations of CD3ε, CD4, CD8α, CD19, CD25, CD44, CD69, γδ TCR, and TCRβ. For human lymphocytes, gating setup was singlets by FSCa and FSCw, lymphocytes by SSCa and FSCa, EMA$^-$, and on subsets as indicated.

## Amino acid radiolabeling

The following reagents were used for radioactive amino acid labeling: Dulbecco's modified Eagle medium (for labeling HeLa cells) or RPMI minus leucine (RPMI without L-leucine, L-glutamine, and sodium pyruvate from MP Biomedicals, supplemented with Glutamax and 1 mM sodium pyruvate) for labeling human lymphocytes with or without inhibitors and for the [$^3$H]-Leu (Perkin Elmer) ribosome transit analysis. Cells were kept at 37°C throughout the experiment and labeling. Cells were resuspended in complete RPMI at $1 \times 10^7$ cells/ml and 1 ml was aliquoted into fresh Eppendorf tubes. Next, cells were spun at $300 \times g$ for 4 min and pre-treated with protein synthesis inhibitors (concentrations as in RPM Staining above) in complete RPMI for 15 min. Pre-treated cells were spun at $300 \times g$ for 4 min, resuspended in 200 µl of labeling media (RPMI-Leu) containing 0.2 mCi/ml [$^3$H]-Leu in the absence or presence of protein synthesis inhibitors as indicated. After a 5-min labeling period, protein synthesis was stopped by adding 1 ml of ice-cold PBS containing 200 µg/ml of CHX. For all labeling experiments, after washing cells thoroughly in ice-cold PBS, cells were lysed in 100 or 200 µl of 2% SDS lysis buffer (2% SDS, 50 mM Tris–HCl pH 7.5, 5 mM ethylenediaminetetraacetic acid (EDTA), 15 U/ml DnaseI (Roche), cOmplete mini EDTA-free protease inhibitor tablet (Roche) in water) and boiled for 30–60 min to ensure complete lysis. Protein amounts were quantified by the DC Protein Assay

(Bio-Rad) or by tryptophan fluorescence measurements (*Wiśniewski and Gaugaz, 2015*). To quantify the amount of [$^3$H]-Leu incorporated into proteins, equal amounts of lysate (six replicates per condition) were spotted onto a 96-well DEAE filter mat (PerkinElmer) and the mat was dried at 60°C. The mat was then soaked in a 10% trichloroacetic (Calbiochem) solution for 30 min at room temperature, washed twice in 70% ethanol, dried at 60°C, placed in a MicroBeta sample bag (PerkinElmer) with ~6 ml of BetaPlate Scint (PerkinElmer), and heat sealed. Radioactivity was quantified in a 1450 MicroBeta TriLux scintillation counter. To determine the total amount of amino acid incorporated into proteins, dilutions of the [$^3$H]-Leu stock were counted and used as standards.

## In vitro RPM-RTA

RPM-RTA: For each time point, $1 \times 10^6$ lymphocytes were transferred to a fresh conical tube and resuspended in 250 µl of the appropriate media. Cells were kept at 37°C (or 39.5°C when indicated) throughout the experiment. An equal volume of 2× inhibitor media was added to each tube at the indicated time and the tube was vortexed briefly to mix. Depending on the time course, the 2× inhibitor media contained HAR (Santa Cruz Biotechnology) at 10 µg/ml (final concentration 5 µg/ml), or HAR at 10 µg/ml and EME (Calbiochem) at 50 µg/ml final concentrations of 5 and 25 µg/ml, respectively. At the end of the time course, an equal volume (250 µl) of 3× PMY (Calbiochem) media (150 µg/ml, for a final concentration of 50 µg/ml) was added to each tube and the tube was vortexed briefly to mix. Cells were incubated for 5 min with PMY before adding an excess of ice-cold PBS to quench the ribopuromycylation reaction. The cells were then stained for analysis by flow cytometry as described above.

[$^3$H]-Leu RTA: Cells were kept at 37°C throughout the experiment. For each time point, $30 \times 10^6$ lymphocytes were transferred to fresh Eppendorf tubes and resuspended in 50 µl of the RPMI-Leu. 50 µl of 2× inhibitor media was added to each tube and the tubes were vortexed briefly to mix. The inhibitor concentrations are as noted above. At the indicated times, an equal volume (100 µl) of [$^3$H]-Leu labeling media (RPMI-Leu media and [$^3$H]-Leu in a 1:1 ratio, 0.5 mCi/ml) was added and cells were labeled for 5 min. To stop [$^3$H]-Leu incorporation, an excess of ice-cold PBS containing 200 µg/ml CHX and 1 mg/ml leucine was added before placing the cells on ice. Cells were lysed and [$^3$H]-Leu incorporation quantified as described under 'Amino acid radiolabeling' subsection.

## Polysome profiling

A 15–45% continuous sucrose gradient was made in Thinwall polyallomer tubes (Beckman Coulter) from 15% and 45% sucrose (MP Biomedicals) solutions in gradient buffer (20 mM Tris–HCl pH 7.4, 5 mM MgCl$_2$, 100 mM KCl, supplemented with 100 µg/ml CHX (Sigma) and 10 U/ml RNaseOUT (Invitrogen)). Briefly, 5 ml of the 15% sucrose solution was carefully layered onto 5 ml of the 45% sucrose solution, and the tube was placed horizontally at 4°C, typically overnight or for at least 2.5 hr before the experiment. For each gradient, cells were harvested and washed in cold PBS as described above. For cell lysis, cells were first swelled by adding 950 µl of a cold hypotonic buffer (20 mM Tris–HCl pH 7.4, 5 mM MgCl$_2$, 10 mM KCl, 40 U/ml RNaseOUT, 0.1 U/µl SuperaseIn, supplemented with Complete EDTA-free protease inhibitors (Roche)). After 10 min of cell swelling, NP-40 was added to a final concentration of 0.5%, the resultant lysate mixed, incubated on ice for 3 min, and spun at 7000 rpm for 2 min to remove nuclei. Post-nuclear lysates were then brought to 100 mM KCl (or 300 or 500 mM NaCl where indicated), layered onto 15–45% continuous sucrose gradients, and spun for 100 min at 38,000 rpm at 4°C in a SW41Ti rotor (Beckman Ultracentrifuge). Gradients were syringe-fractionated mechanically from the bottom up and monitored for absorbance at 254 nm (Teledyne Isco) to obtain polysome profiles. When indicated, area under the curve measurements were calculated by a trapezoidal method from the resulting curves. When required by the experiment, we concentrated monosome and polysome fractions for immunoblotting. To pellet the ribosomes, we spun the pooled monosome or polysome fractions for 1 hr or O/N at 39,000 rpm at 4°C in the SW41Ti rotor on a 34% sucrose cushion. The resultant ribosome pellet was resuspended in 2% SDS extraction buffer.

## Quantitating cellular proteins

We quantitated total protein in cell lysates based on Trp fluorescence (*Wiśniewski and Gaugaz, 2015*). Briefly, cells (1–2 million lymphocytes per 100 µl) were lysed for 10–30 min at 95°C in a solution of 2% SDS, 0.1 M Tris–HCl, 50 mM Dithiothreitol (DTT), pH 7.8 with 15 U/ml DnaseI (Roche) and a

cOmplete mini EDTA-free protease inhibitor tablet (Roche) added fresh. An 8 M urea, 10 mM Tris–HCl with 0.5 mM DTT solution was freshly prepared, and 200 µl added to wells of a flat-bottomed black polystyrene plate, and 2–4 µl of either cell lysates or a Trp standard solution was added to individual wells in triplicate. Fluorescence emitted at 350 nm after excitation at 295 nm was measured. We also compared this assay with the commercially available DC protein assay (Bio-Rad, performed according to the manufacturer's instructions), and found that the assays generated similar values.

## Quantitating cellular ribosomes

After lymphocyte isolation, the PBS-washed cell pellet was dissolved in TRIzol; a spike-in mRNA standard was added at this step (CleanCap EGFP mRNA from TriLink, L-7601) to account for RNA loss during processing. RNA purification was conducted as described in the manufacturer's TRIzol protocol, with 5 µg of glycogen used as carrier and the isopropanol precipitation step conducted overnight at −20°C. The final RNA pellet was dissolved in 50 µl of ultra-pure water and roughly quantified to determine appropriate range for the Agilent bioanalyzer chip. Samples, including fresh spike-in mRNA alone, were loaded and run on RNA Nano Bioanalyzer chips (Agilent RNA Nano 6000), with a 70°C heating step and run on a 2100 Agilent Bioanalyzer. Bioanalyzer 2100 Expert software was used to determine total RNA concentration of each sample and percent area under curve of each peak (mRNA spike-in at ~1000 bp, 18S rRNA at ~1800 bp, 28S rRNA at ~4000 bp). The yield of the RNA prep was calculated as follows:

(mRNA spike-in standard peak from an RNA-purified sample)/average of 2–3 standard peaks from 75 ng/µl standard wells) = (fraction of RNA that remains after the purification process). We next converted the 18S and 28S ng/µl values to 'number molecules per cell' using the number of cells that originally went into the RNA purification, the RNA yield described above, and the following values: mouse 18S = 6.40E + 05 g/mol; mouse 28S = 1.60E + 06 g/mol; human 18S = 6.40E + 05 g/mol; human 28S = 1.70E + 06 g/mol.

## Immunoblotting

To fractionate cells into nuclear and non-nuclear lysates, cells were either dissolved directly in 2% SDS extraction buffer at 95°C ('all' in *Figure 6—figure supplement 1*) or subjected to a hypotonic lysis procedure. Cells were swelled with a buffer containing 20 mM Tris–HCl pH 7.4, 2.5 mM MgCl$_2$, and 10 mM KCl supplemented with protease inhibitors for 10 min on ice. NP-40 was added to a final concentration of 0.5%, and the resultant lysate was mixed, incubated on ice for 3 min, and spun at 7000 rpm for 1 min. Non-nuclear lysates were removed and immediately dissolved in gel loading sample buffer (Life Technologies) to prevent sample degradation. Nuclei were washed gently 2× with PBS buffer containing NP-40 and protease inhibitors. Finally, nuclear proteins were extracted by dissolving pelleted nuclei in 2% SDS extraction buffer at 95°C. Equal amounts of each fraction were prepared for SDS–PAGE.

Samples were electrophoresed in 4–12% NuPAGE Bis-Tris gels (Invitrogen). Proteins were then transferred to nitrocellulose membranes (iBlot system, Novex) and membranes stained with Ponceau S and washed with PBS to confirm transfer uniformity. Next, membranes were incubated with either StartingBlock buffer (Thermo Scientific) or Odyssey Blocking Buffer (Licor), followed by primary antibody prepared in StartingBlock buffer or Odyssey Blocking Buffer with 0.1% Tween-20 (Sigma). Depending on the experiment, we used the following primary antibodies: mouse anti-PMY (clone 2A4) at 6.66 µg/ml; human anti-ribosomal P antigen (Immunovision) at 1:2000; rabbit anti-RPL7 (Abcam) at 1:1000, rabbit anti-RPL26 (Bethyl Laboratories) at 1:2000; mouse anti-beta actin (Invitrogen) at 1:4000; rabbit anti-HSP90 (Santa Cruz) at 1:500; rat anti-GRP94 at 1:1500 (Enzo); rabbit anti-RPL28 (Abcam) at 1:500, rabbit anti-RPL6 (Abcam) at 1:1000, mouse anti-PDI at 1: 2000 (Abcam); mouse anti-lamin A/C (Cell Signaling Technology) at 1: 2000, rabbit anti-fibrillarin (Cell Signaling Technology) at 1:1000, mouse anti-RPS6 (Cell Signaling Technology) at 1:1000, rabbit anti-histone H3 (Cell Signaling Technology) at 1: 2000, rabbit anti-RPL5 (Cell Signaling Technology) at 1:1000, and rabbit anti-RPS3 (Cell Signaling Technology) at 1:1000 (Cell Signaling Technology). The number of ribosomes per cell in early optimization experiments was quantified by generating a standard curve using highly purified HeLa cell or canine rough microsome ribosomes (a kind gift of Chris Nicchitta, Duke University).

Membranes were washed three times in PBS + 0.1% Tween-20 (PBS-T) followed by incubation with secondary antibodies (all from Licor; used at 1:10,000) prepared in StartingBlock buffer or Odyssey

Blocking Buffer. Membranes were washed 3× in PBS-T, 1× in PBS, and scanned via a Licor Odyssey CLX scanner.

## Acknowledgements

We are grateful to Chris Nicchitta (Duke University) for purified ribosomes and Glennys Reynoso for outstanding technical assistance. This work was supported by the Division of Intramural Research, National Institute of Allergy and Infectious Diseases.

## Additional information

### Funding

| Funder | Grant reference number | Author |
| --- | --- | --- |
| Division of Intramural Research, National Institute of Allergy and Infectious Diseases | | Jonathan W Yewdell |

The funders had no role in study design, data collection, and interpretation, or the decision to submit the work for publication.

### Author contributions

Mina O Seedhom, Devin Dersh, Conceptualization, Data curation, Formal analysis, Validation, Investigation, Visualization, Methodology, Writing – original draft, Writing – review and editing; Jaroslav Holly, Data curation; Mariana Pavon-Eternod, Conceptualization, Data curation, Formal analysis, Validation, Investigation, Methodology; Jiajie Wei, Jefferson Santos, Data curation, Investigation; Matthew Angel, Data curation, Formal analysis, Investigation; Lucas Shores, Formal analysis, Investigation; Alexandre David, Conceptualization, Supervision, Investigation; Heather Hickman, Data curation, Formal analysis, Validation, Investigation; Jonathan W Yewdell, Conceptualization, Formal analysis, Supervision, Funding acquisition, Investigation, Methodology, Writing – original draft, Project administration, Writing – review and editing

### Author ORCIDs

Mina O Seedhom ⓘ http://orcid.org/0000-0003-2123-6189
Devin Dersh ⓘ http://orcid.org/0000-0002-3763-4192
Jefferson Santos ⓘ http://orcid.org/0000-0001-5895-4163
Jonathan W Yewdell ⓘ https://orcid.org/0000-0002-3826-1906

### Ethics

All animal studies were approved by and performed in accordance with the Animal Care and Use Committee of the National Institute of Allergy and Infectious Diseases under protocol LVD-5E.

Reviewer #1 (Public Review): https://doi.org/10.7554/eLife.89015.3.sa1
Reviewer #2 (Public Review): https://doi.org/10.7554/eLife.89015.3.sa2
Reviewer #3 (Public Review): https://doi.org/10.7554/eLife.89015.3.sa3
Author Response https://doi.org/10.7554/eLife.89015.3.sa4

## Additional files

### Supplementary files
• MDAR checklist

### Data availability

All data generated or analyzed during this study are included in the manuscript and supporting files. Numerical data, statistics, and uncropped/unedited immunoblot images are found in source data files.

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
