## [Editor Report · eLife assessment]

This study addresses how protein synthesis in activated lymphocytes keeps up with their rapid division, with **important** findings that are of significance to cell biologists and immunologists endeavouring to understand the 'economy' of the immune system. The work is supported by **solid** data. Because it proposes non-conventional mechanisms, the study sets the scene for further work in this area.

---

## [Referee Report · Reviewer #1 (Public Review)]

The authors examine the fascinating question of how T lymphocytes regulate proteome expression during the dramatic cell state change that accompanies the transition from the resting quiescent state to the activated, dividing state. Orthogonal, complementary assays for translation (RPM/RTA, metabolic labeling) are combined with polyribosome profiling and quantitative, biochemical determinations of protein and ribosome content to explore this question, primarily in the OT-I T lymphocyte model system. The authors conclude that the ratio of protein levels to ribosomes/protein synthesis capacity is insufficient to support activation-coupled T cell division and cell size expansion. The authors hint at cellular mechanisms to explain this apparent paradox, focusing on protein acquisition strategies, including emperipolesis and entosis, though these remain topic areas for future study.

The strengths of the paper include the focus on a fundamental biological question - the transcriptional/translational control mechanisms that support the rapid, dramatic cell state change that accompanies lymphocyte activation from the quiescent to activated state, the use of orthogonal approaches to validate the primary findings, and the creative proposal for how this state change is achieved.

The weakness of the work is that several cellular regulatory processes that could explain the apparent paradox are not explored, though they are accessible to experimental analysis. In the accounting narrative that the authors highlight, a thorough accounting of the cellular process inventory that could support the cell state change should be further explored before committing to the proposal, provocative as it is, that protein acquisition provides a principal mechanism for supporting lymphocyte activation cell state change.

Appraisal and Discussion:

1. Relating to the points raised above, two recent review articles explore this topic area and highlight important areas of study in RNA biology and translational control that likely contribute to the paradox noted by the authors: Choi et al. 2022,

2. The authors cite the Wolf et al. study from the Geiger lab (doi.org/10.1038/s41590-020-0714-5, ref. 41) though largely to compare determined values for ribosome number. Many other elements of the Wolf paper seem quite relevant, for example, the very high abundance of glycolytic enzymes (and whose mRNAs are quite abundant as well), where (and as others have reported) there is a dramatic activation of glycolytic flux upon T cell activation that is largely independent of transcription and translation, the evidence for "pre-existing, idle ribosomes", the changes in mRNA copy number and protein synthesis rate Spearman correlation that accompanies activation, and that the efficiencies of mRNA translation are heterogeneous. These data suggest that more accounting needs to be done to establish that there is a paradox.

As one example, what if glycolytic enzyme protein levels in the resting cell are in substantial excess of what's need to support glycolysis (likely true) and so translational upregulation can be directed to other mRNAs whose products are necessary for function of the activated cell? In this scenario the dilution of glycolytic enzyme concentration that would come with cell division would not necessarily have a functional consequence. And the idle ribosomes could be recruited to key subsets of mRNAs (transcriptionally or post-transcriptionally upregulated) and with that a substantial remodeling of the proteome (authors ref. 44). The study of Ricciardi et al. 2018 (The translational machinery of human CD4+ T cells is poised for activation and controls the switch from quiescence to metabolic remodeling [doi.org/10.1016/j.cmet.2018.08.009]) is consistent with this possibility. That study, and the short reviews noted above, are useful in highlighting the contributions of selective translational remodeling and the signaling pathways that contribute to the cell state change of T cell activation. From this perspective an alternative view can be posited, where the quiescent state is biologically poised to support activation, where subsets of proteins and mRNAs are present in far higher levels than that necessary to support basal function of the quiescent lymphocyte. In such a model, the early stages of lymphocyte activation and cell division are supported by this surplus inventory, with transcriptional activation, including ribosomal genes, primarily contributing at later stages of the activation process. An obvious analogy is the developing *Drosophila* embryo where maternal inheritance supports early-stage development and zygotic transcriptional contributions subsequently assuming primary control (e.g. DOI 10.1002/1873-3468.13183, DOI: 10.1126/science.abq4835). To pursue that biological logic would require quantifying individual mRNAs and their ribosome loading states, mRNA-specific elongation rates, existing individual protein levels, turnover rates of both mRNAs and proteins, ribosome levels, mean ribosome occupancy state, and how each of these parameters are altered in response to activation. Such accounting could go far to unveil the paradox. This is a considerable undertaking, though, and outside the scope of the current paper.

Regarding the revised manuscript:

I am largely satisfied with the authors responses to the review and have but a few remaining thoughts, some mirrored in the comments from the other reviewers and some that came to mind upon reading the revision.

1. In the Introduction, it would be (have been) helpful if in paragraph two, it was stated that the current study was designed to test that assumption made in prior reports that the fold-increase in protein synthesis in response to mitogen activation was sufficient to endow the daughter cells with "the same protein content as their progenitor".

2. The primary conclusion, that "...protein synthesis activity or capacity of in vivo activated T cells does not support their doubling times" remains, to my eye, insufficiently supported by the data, though I agree it is a rational interpretation. My concern is that the devil is deeper in the details and without knowing the mRNA transcriptome composition pre- and post-activation, mean CDS length, 5' UTR structural features, perhaps codon optimality, etc., etc., the broader conclusion could be premature. As a first check, it would be useful to determine poly(A) mRNA and ribosome concentrations/cell, pre- and post-activation, and subsequently to compare mRNA transcriptome compositions in greater detail. Do mRNA:ribosome levels and ratios diverge as a consequence of activation? Poly(A) mRNA compositions? Does protein half-life change pre- and post-activation? mRNA half-life? My view is that additional molecular accounting is likely necessary to be confident in the primary conclusion.

3. I did not provide a clear description of the alternative interpretation I was imagining, which is that in the resting, unstimulated state, mRNA:ribosome and/or protein levels may be much higher than that necessary for lymphocyte viability. As in early development, this could be a mechanism to then provide sufficient protein synthesis capacity and/or proteins to daughter cells following activation of cell division and cell growth. In other words, it's a dynamic range question; the daughter cells exploit "unused" protein synthesis capacity to sustain their growth and division. Quantification and analysis of the additional variables noted in point (2) could reconcile the different interpretations.

---

## [Referee Report · Reviewer #2 (Public Review)]

This paper takes a novel look at the protein economy of primary human and mouse T-cells - in both resting and activated state. Their findings in primary human T-cells are that:

1. A large fraction of ribosomes are stalled in resting cultured primary human lymphocytes. and these stalled ribosomes are likely to be monosomes.

2. Elongation occurs at similar rates for HeLa cells and lymphocytes, with the active ribosomes in resting lymphocytes translating at a similar rate as fully activated lymphocytes.

They then turn their attention to mouse OT-1 lymphocytes, looking at translation rates both in vitro and in vivo. Day1 resting T-cells also show stalling - which curiously wasn't seen on freshly purified cells - I didn't understand these differences.

In vivo they show that it is possible to monitor accurate translation and to measure rates in vivo. Perhaps most interestingly they note a paradoxically high ratio of cellular protein to ribosomes insufficient to support their rapid in vivo division, suggesting that the activated lymphocyte proteome in vivo may be generated in an unusual manner.

This was an interesting and provocative paper. Lots of interesting techniques and throwing down challenges to the community - it manages to address a number of important issues without necessarily providing answers.

---

## [Referee Report · Reviewer #3 (Public Review)]

Perhaps not unexpectedly, the proposed revisions consist of textual revisions only. Yewdell added a touch of levity with his H.G. Wells foundation as a source of $$ for a time machine. The paper does not establish striking new facts, in my opinion, but will stimulate discussion.

One point to consider: the relevance of the human T cell activation experiments is now downplayed even further, by the authors themselves, no less. I would suggest leaving the actual data out altogether and conclude with a statement: "Similar experiments conducted on activated human T cells showed significantly worse activation and may therefore not allow a head-to-head comparison with the results of our experimentst performed on mouse T cells. Not only might one consider the mode of activation (PMA/ionomycin) non-physiological, the activation status achievedwas less than that seen for the OT-1 model. " or something similar to that effect. In the present weakened form, I do not believe that the human data add anything of substance to the paper and are more of a distraction. The authors would increase the impact and readability of their paper if they omitted the human data.

---

## [Author Response]

The following is the authors’ response to the original reviews.

**eLife assessment**
This study addresses how protein synthesis in activated lymphocytes keeps up with their rapid division, with important findings that are of significance to cell biologists and immunologists endeavouring to understand the 'economy' of the immune system. The work is supported by solid data but because it proposes non-conventional mechanisms, it requires additional explanation and justification to align with the current understanding in the field.
**Public Reviews:**

**Reviewer #1 (Public Review):**
The authors examine the fascinating question of how T lymphocytes regulate proteome expression during the dramatic cell state change that accompanies the transition from the resting quiescent state to the activated, dividing state. Orthogonal, complementary assays for translation (RPM/RTA, metabolic labeling) are combined with polyribosome profiling and quantitative, biochemical determinations of protein and ribosome content to explore this question, primarily in the OT-I T lymphocyte model system. The authors conclude that the ratio of protein levels to ribosomes/protein synthesis capacity is insufficient to support activation-coupled T cell division and cell size expansion. The authors hint at cellular mechanisms to explain this apparent paradox, focusing on protein acquisition strategies, including emperipolesis and entosis, though these remain topic areas for future study.The strengths of the paper include the focus on a fundamental biological question - the transcriptional/translational control mechanisms that support the rapid, dramatic cell state change that accompanies lymphocyte activation from the quiescent to activated state, the use of orthogonal approaches to validate the primary findings, and the creative proposal for how this state change is achieved.The weakness of the work is that several cellular regulatory processes that could explain the apparent paradox are not explored, though they are accessible for experimental analysis. In the accounting narrative that the authors highlight, a thorough accounting of the cellular process inventory that could support the cell state change should be further explored before committing to the proposal, provocative as it is, that protein acquisition provides a principal mechanism for supporting lymphocyte activation cell state change.Appraisal and Discussion:1. relating to the points raised above, two recent review articles explore this topic area and highlight important areas of study in RNA biology and translational control that likely contribute to the paradox noted by the authors: Choi et al. 2022, doi.org/10.4110/in.2022.22.e39 ("RNA metabolism in T lymphocytes") and Turner 2023, DOI: 10.1002/bies.202200236 ("Regulation and function of poised mRNAs in lymphocytes"). These should be cited, and the broader areas of RNA biology discussed by these authors integrated into the current manuscript.

Good suggestion. We have added these references with a short discussion.

2. The authors cite the Wolf et al. study from the Geiger lab (doi.org/10.1038/s41590-020-07145, ref. 41) though largely to compare determined values for ribosome number. Many other elements of the Wolf paper seem quite relevant, for example, the very high abundance of glycolytic enzymes (and whose mRNAs are quite abundant as well), where (and as others have reported) there is a dramatic activation of glycolytic flux upon T cell activation that is largely independent of transcription and translation, the evidence for "pre-existing, idle ribosomes", the changes in mRNA copy number and protein synthesis rate Spearman correlation that accompanies activation, and that the efficiencies of mRNA translation are heterogeneous. These data suggest that more accounting needs to be done to establish that there is a paradox.As one example, what if glycolytic enzyme protein levels in the resting cell are in substantial excess of what's needed to support glycolysis (likely true) and so translational upregulation can be directed to other mRNAs whose products are necessary for function of the activated cell? In this scenario, the dilution of glycolytic enzyme concentration that would come with cell division would not necessarily have a functional consequence. And the idle ribosomes could be recruited to key subsets of mRNAs (transcriptionally or post-transcriptionally upregulated) and with that a substantial remodeling of the proteome (authors ref. 44). The study of Ricciardi et al. 2018 (The translational machinery of human CD4+ T cells is poised for activation and controls the switch from quiescence to metabolic remodeling (doi.org/10.1016/j.cmet.2018.08.009) is consistent with this possibility. That study, and the short reviews noted above, are useful in highlighting the contributions of selective translational remodeling and the signaling pathways that contribute to the cell state change of T cell activation.

Our study focuses on the central issue of whether measured ribosome translation rates support rapid division. The abundance of glycolytic enzymes, mRNA copy numbers etc., are clearly interesting and critical to cell metabolism, but are irrelevant to measuring the overall translation rate and capacity of T cells.

From this perspective, an alternative view can be posited, where the quiescent state is biologically poised to support activation, where subsets of proteins and mRNAs are present in far higher levels than that necessary to support basal function of the quiescent lymphocyte. In such a model, the early stages of lymphocyte activation and cell division are supported by this surplus inventory, with transcriptional activation, including ribosomal genes, primarily contributing at later stages of the activation process. An obvious analogy is the developing *Drosophila* embryo where maternal inheritance supports early-stage development and zygotic transcriptional contributions subsequently assuming primary control (e.g. DOI 10.1002/1873-3468.13183 , DOI: 10.1126/science.abq4835). To pursue that biological logic would require quantifying individual mRNAs and their ribosome loading states, mRNA-specific elongation rates, existing individual protein levels, turnover rates of both mRNAs and proteins, ribosome levels, mean ribosome occupancy state, and how each of these parameters is altered in response to activation. Such accounting could go far to unveil the paradox. This is a considerable undertaking, though, and outside the scope of the current paper.

The reviewer is essentially proposing RiboSeq analysis of pre- and post-activation T cells, whereby individual mRNAs can be queried for ribosome occupancy, and where translation inhibitors could be used to quantify mRNA-specific transit rates. This is important information but would not provide a more accurate accounting of protein synthesis rates than our much more direct measurement. We note that other labs have begun to work on this exact topic, however – see both PMID: 36002234 and PMID: 32330465.

**Reviewer #2 (Public Review):**
This paper takes a novel look at the protein economy of primary human and mouse T-cells - in both resting and activated state. Their findings in primary human T-cells are that:1. A large fraction of ribosomes are stalled in resting cultured primary human lymphocytes, and these stalled ribosomes are likely to be monosomes.2. Elongation occurs at similar rates for HeLa cells and lymphocytes, with the active ribosomes in resting lymphocytes translating at a similar rate as fully activated lymphocytes.They then turn their attention to mouse OT-1 lymphocytes, looking at translation rates both in vitro and in vivo. Day 1 resting T-cells also show stalling - which curiously wasn't seen on freshly purified cells - I didn't understand these differences.

This is clarified and discussed starting in the third paragraph of “Protein synthesis in mouse lymphocytes ex vivo” section. Cells cultured ex vivo for 1 day with no activation show signs of stalling, as we observed in isolated human cells. But cells immediately out of an animal show a measurable decay rate since they are obviously synthesizing proteins in vivo and are processed rapidly.

In vivo, they show that it is possible to monitor accurate translation and measure rates. Perhaps most interestingly they note a paradoxically high ratio of cellular protein to ribosomes insufficient to support their rapid in vivo division, suggesting that the activated lymphocyte proteome in vivo may be generated in an unusual manner.This was an interesting and provocative paper. Lots of interesting techniques and throwing down challenges to the community - it manages to address a number of important issues without necessarily providing answers.
**Reviewer #3 (Public Review):**
This manuscript provides a more or less quantitative analysis of protein synthesis in lymphocytes. I have no issue with the data as presented, as I'm sure all measurements have been expertly done. I see no need for additional experimental work, although it would be helpful if the authors could comment on the possibility of measuring the rate of synthesis of a defined protein, say a histone, in cells prior to and after activation. The conclusion the authors leave us with is the idea that the rates of protein synthesis recorded here are incompatible with observed rates of T cell division in vivo. Indeed, in the final paragraph of the discussion, the authors note the mismatch between what they consider a requirement for cell division, and the observed rates of protein synthesis. They then invoke unconventional mechanisms to make up for the shortfall, without -in this reviewer's opinion- discussing in adequate detail the technical limitations of the methodology used.

Points #1-3 in the Discussion relate to potential pitfalls of our analyses; in point #3 we now add further limitations of RTA based on non-random detection of nascent chains due either to bias in either puromycylation or antibody detection of puromycylated nascent chains.

A key question is the broad interest, novelty, and extension of current knowledge, in comparison with Argüello's (reference 27) 'SunRise' method. It would be helpful for the authors to stake out a clear position as to the similarities and differences with reference 27: what have we learned that is new? The authors could cite reference 27 in the introduction of their manuscript, given the similarity in approach. That said, the findings reported here will generate further discussion.

We did cite this reference (27) in the section “Flow RPM measures ribosome elongation rate in live cells” giving credit where credit is due. We independently devised the method in 2014, and uniquely, to our knowledge, have applied it in vivo. We now further discuss the importance of our CHX modification to limit dissociation and increase the accuracy of RTA (second and third paragraphs of “Protein synthesis in mouse lymphocytes and innate immune cells in vivo”).

The manuscript would increase in impact if the authors were to clearly define why a particular measurement is important and then show the actual experiment/result. As an example, it would be helpful to explain to the non-expert why the distinction between monosomes, polysomes, and stalled versions of the same is important, and then explain the rationale of the actual experiment: how can these distinctions be made with confidence, and what are confounding variables?

We believe this is addressed in the section “Resting human lymphocytes have a dominant monosome population”.

The initial use of human cells, later abandoned in favor of the OT-1 in vitro and in vivo models, requires contextualization. If the goal is to address the relationship between rates of translation and cell division of antigen-activated T cells in vivo, then a lot of the work on the human model and the in vitro experiments becomes more of a distraction, unless properly contextualized. Is there any reason to assume that antigen-specific activation in vivo will impact translation differently than the use of the PMA/ionomycin/IL2 cocktail? The way the work is presented leaves me with the impression that everything that was done is included, regardless of whether it goes to the core of the question(s) of interest.

Donor PBMCs are clearly the more relevant model for understanding human T cell biology, which is why started our studies with this model. Had the manuscript strictly described mouse studies it is likely that we would be criticized for not studying human cells: Catch 22! However, as we state in the manuscript, the human cell model has a variety of technical downsides, including donor heterogeneity. PMA/ionomycin activation is also physiologically questionable, and while we could deliver a defined TCR to redirect their specificity, this is typically done after cells have been activated, since lentiviral delivery is poor in resting lymphocytes. A main point we try to make from this work is that cells derived from human blood donors show signs of ribosomal stalling by the time they are isolated and put into culture. This may limit the usefulness of studying them preactivation, although based on our mouse data, some level of stalled ribosomes may be a feature as well – to prime T cells to be ready for their massive expansion. The move to the OT-I system gave us complete control over the system, including in vivo delivery of translation inhibitors.

It would be helpful if the authors made explicit some of the assumptions that underlie their quantitative comparisons. Likewise, the authors should discuss the limitations of their methods and provide alternative interpretations where possible, even if they consider them less/not plausible, with justification. As they themselves note, improvements in the RPM protocols raised the increase in translating ribosomes upon activation from 10-fold to 15-fold. Who's to say that is the best achievable result? What about the reliability/optimization of the other measurements?

We expanded discussion of potential pitfalls of the RPM techniques and others in the Discussion section. Regarding RPM per se, we use it as a readout of ribosome time decay, so even if further optimizations can be made, the decay rates we have made should still be accurate. In addition, for our cell accounting measurements in Figure 6, we do not use RPM data and rather calculate based on the assumption that every ribosome is used for protein synthesis at a “maximal” rate of mRNA transit.

The composition of the set of proteins produced upon activation will differ from cell to cell (CD4, CD8, B, resting vs. dividing). Even if analyses are performed on fixed cells, the ability of the monoclonal anti-puromycin antibody to penetrate the matrix of the various fixed cell types may not be equal for all of them, depending on protein composition, susceptibility to fixation etc. Is it possible for puromycin to occupy the ribosome's A site and terminate translation without forming a covalent bond with the nascent chain? This could affect the staining with anti-puromycin antibodies and also underestimate the number of nascent chains.

Yes, the method (like every other one) is imperfect. Harringtonine run-off experiments show that RPM staining only detects nascent chains. Note that reference 47 reports that 75% of translation in activated T cells is devoted to synthesizing ~250 housekeeping proteins, which are likely to be highly similar between lymphocyte subsets.

I believe that the concept of FACS-based quantitation also requires an explanation for the nonexpert. For the FACS plots shown, the differences between the highest and lowest RPM scores for cells that divided and that have a similar CFSE score is at least 10-fold. Does that mean that divided cells can differ by that margin in terms of the number of nascent chains present? If I make the assumption that cells stimulated with PMA/ionomycin/IL2 respond more or less synchronously, why would there be a 10-fold difference in absolute fluorescence intensity (anti=puromycin) for randomly chosen cells with similar CFSE values? While the use of MFI values is standard practice in cytofluorimetry, the authors should devote some comments to such variation at the population level.

We believe that the referee is referring to Sup Fig. 1B. In this experiment the T cells are polyclonal and represent the full range of naïve to potentially exhausted differentiation states. Looking at our initial in vivo RPM study (reference 22) and comparing Figure 2 (OTI’s) to Figure 3 (endogenous CD4s or CD8s), reveals more spread in the RPM values polyclonal vs. monoclonal T cells - now clarified in the third paragraph of “Protein synthesis in mouse lymphocytes and innate immune cells in vivo”. Flow cytometry is by far the most accurate method for measuring fluorescence in individual cells. It is likely to be an accurate measure of the variation of nascent chains in cells in the same division cohort but likely represents the diversity of T cell activation profiles in blood of healthy donors.

It is assumed that for cells to complete division, they must have produced a full and complete copy of their proteome and only then divide. What if cells can proceed to divide even when expressing a subset of the proteome of departure (=the threshold set required for initiation of division), only to complete synthesis of the 'missing ' portion once cell division is complete? Would this obviate the requirement for an unusual mechanism of protein acquisition (trogocytosis; other)?

There must be a steady state level of translation and proteome replenishment, though. If a cell can divide when it affords daughter cells with 90% of its G0 proteome (as an example), that daughter cell would either (1) be 10% smaller, or (2) require extra translation to make up for the missing proteome during its own division cycle. Though T cells do typically shrink slightly after an initial activation, cell size stabilizes over time. Requiring each daughter cell to make more and more missing proteome could be plausible, considering that initial bursts of division do take longer over time, but still, even in vitro activated T cells divide rapidly for weeks without large decreases in their division rates.

Translation is estimated to proceed at a rate of ~6 amino acids per second, but surely there is variability in this number attributable to inaccuracies of the methods used, in addition to biological variability. Were these so-called standard values determined for a range of different tissues? It stands to reason that there might be variation depending on the availability of initiation/elongation factors, NTPs, aminoacyl tRNAs etc. What is the margin of error in calculating chain elongation rates based on the results shown here?

We refer to all relevant studies we know of, including new in vivo estimates of elongation rates (reference 40).

**Reviewer #1 (Recommendations For The Authors):**
A "limitations of study" section would be a helpful way to detail potential contributing mechanisms that were not explored in the current study.

We have expanded the methodological limitations in the Discussion section.

Major:1. Broaden the scope of biological models that could explain the paradox.

In the Discussion, we suggest that T cells acquire some fraction of their proteome through external sources and highlight some examples of this occurring.

Minor:1. Include Mr markers for Fig. 2C.

Done.

2. Though commonly used interchangeably, historically the term protein synthesis was the consequence of mRNA translation. In other words, proteins are not translated.

Good point! We have changed the text accordingly.

3. Include more meaningful X-axis legend in polysome gradient panels i.e., Fig. S2, e.g., fraction number.

In most experiments, fractions were not collected. Rather, the x-axis refers to time that the sample took to be queried by the detector.

4. Figure 3A does not report polysome profiles as described in the text, pg. 5, though this is reported in Fig S2D.

The figure callouts were correct but confusing. We now separately refer to out each result to clarify.

5. In Fig 5A, SDS-PAGE/anti-Puro blots would be more convincing and contain more information. The dot-blot is difficult to interpret.

Disagree. To quantitate total anti-puromycin signal a dot blot is far better than immunoblotting, which is compromised by unequal transfer of different protein species.

6. It's not clear why a degree of monosome translation is necessarily surprising (pg. 7).

It’s surprising since for many decades it was believed that translation by monosomes is a tiny fraction of translation. But separately, with this particular mode of activation, activated T cells displayed a preponderance of monosomes during their burst of division. When the activation method was improved, polysomes dominated. But monosome translation clearly supported T cell division during activation without cognate peptide, which was interesting.

**Reviewer #2 (Recommendations For The Authors):**
1. One concern is the dose of puromycin used. My understanding is that puromycin acts as a chain termination inhibitor - but is being used here predominantly as a label for nascent polypeptide chains. My concern, therefore, is the dose being used - here at 50ug/ml - which seems high and I would be concerned that at this dose it would act as a translational inhibitor rather than just labelling nascent chains, and is therefore resulting in a lower signal/background ration than expected. In human cell lines 0.1ug/ml is optimal and doses published (in cell lines) range between 1 and 10ug/ml so it will be interesting to understand why this high dose was used.Do they have a dose-response curve - is this high dose necessary because these are primary Tcells. Can the authors show that 50 µg/mL of puromycin is optimal for studying protein translation in primary human T cells? A titration curve will help answer this question and could be included in Suppl Figure 1. This experiment is critical as the authors use a higher dose than previous studies (commonly between 1 and 10 µg/mL).

The reviewer is referencing puromycin concentrations typically used in the selection of cells – for the RPM assay, puromycin is used at saturating doses to label the maximal number of nascent chains stalled by CHX or EME pretreatment.

2. None of the figures show statistical significance.

Statistics on relevant comparisons are now indicated on figures and in legends.

3. The authors mention: "We performed RPM on cells labelled with CFSE to track cell division by dye dilution (Supplemental Figure 1B). On day 2, activated cells exhibited multiple populations, with nearly all divided cells showing a high RPM signal.". However, on day 2 it is hard to see any dividing cells in the dot plot included in the supplemental figure. Dividing cells only appear on day 5? Their statements make the subsequent paragraphs also difficult to follow.

We modified the text to clarify this data – there is likely activation-induced cell death occurring which is why there are relatively few CFSE-low cells at this timepoint, and they do exhibit a fairly wide range of RPM staining. The main point is that by day 5, nearly all divided cells exhibit high RPM.

4. "Many divided cells exhibited near baseline RPM signals, however, consistent with their return to the resting state. Interestingly, although non-activated cells did not divide, ~50% demonstrated increased RPM staining.". Again, it is hard to see the ~50% of cells with increased RPM the authors refer to in the provided supplemental figure.

This is from quantification of the flow data and is described more fully later when we discuss ribosome stalling.

5. The authors say "Thus, we cannot attribute the persistence of flow RPM staining in translation initiation inhibitor-treated cells to incomplete inhibition of protein synthesis.' - but it's unclear what this refers to as in the previous paragraph they also say: 'Initiation inhibitors, however, clearly discriminated between day 1 resting and activated cells. RPM signal was diminished by up to 8090% on day 5 post-activation.' - this is all somewhat confusing. It would be helpful to have this clarified and in the text to make more liberal use of referring to specific figures.

Figure 1B shows that RPM is maintained at fairly high levels during treatment with EME or CHX (in contrast to the initiation inhibitors HAR/PA). To rule out that the drugs were simply not active, tritiated leucine labeling was conducted to confirm that incorporation of the radiolabeled amino acid dropped to near-baseline (Figure 1C). Therefore, we can conclude that the drugs are indeed working as intended, but EME/CHX does not decrease RPM signal to the same extent that they prevent leucine incorporation.

6. Page 5 Fig 3A - I don't understand the difference between freshly isolated OT-1 cells - which don't stall and day 1 OT-1 cells which do. Why are freshly isolated cells not behaving like the naïve cells- isn't this what they would predict? Also - I accept that there is a move from monosome to polysome population between day 1 and 2 - the effect isn't huge - it would be helpful/interesting to know what has happened by day 5 - is the effect much more significant?

Freshly isolated cells are harvested from animals and immediately queried, whereas day 1 cells are cultured for 24h in the absence of any activation. Presumably, the ex vivo culture without any activation causes the mouse T cell ribosomes to stall, just as we observed in cells obtained from human donors that took hours to collect and bring to the bench. The appearance of polysomes is really related to how the activation of the cells is done… refer to Figure 5B to see how significant the polysome buildup can be!

7. Fig S3C - I don't understand how they reach the conclusion from this figure that: '~15-fold increase in translating ribosomes in activated OT-I T cells in vivo (Supplemental Figure 3C) as compared to the 10-fold increase we previously reported using the original protocol. It would very much help the reader if these calculations could be better explained.

These are simply quantifications of the RPM staining done in Supplemental Figure 3C compared to experiments done in the absence of the CHX-modified method.

8. Page 7 - They conclude that the Tan paper has superior lymphocyte activation - but presumably this depends on the signal as to whether there is more activation and how this affects the shift from monosome to polysome -ie maybe a stronger activation signal affects the distribution more - perhaps their method is the more physiological? Is their conclusion fair - that 'These findings indicate that monosomes make a major contribution to translation in resting T cells but are likely to make a minor contribution in fully activated cells.'

Yes, we believe that their published method would be more physiological with the use of the natural OT-I peptide. We conclude that although monosome translation is present (as others have published), there are relatively few monosomes in fully activated T cells. Therefore, the monosome contribution to overall translation in activated T cells appears to be minor.

9. Contrary to observations in vitro, ribosomes are not stalled in naïve mouse T cells in vivo, as we show via RTA analysis of non-activated T cells. - yes - this seems somewhat surprising - what is the explanation?

We presume this is due to the stress/non-native environment that ex vivo cultured cells are subjected to.

10. Whilst I understand the point that the authors are trying to make in Figure 1D about resting T cells having high background RPM staining due to stalled ribosomes, it is intriguing that there is almost no difference (no statistical significance provided) after 2 or 5 days of activation. Isn't this finding contrary to the one provided in Figure 1A and Suppl Figure 1B?

Figure 1A is showing the difference between no activation and activation conditions. Figure 1D is predominantly meant to show that the increase in RPM from activated cells at day 1 and day 5 are not as different as one might predict. The reason, as we describe in further experiments, is likely that cells exhibiting ribosomal stalling can incorporate puromycin, damping the “fold change” we calculate (unlike what we observe in metabolic labeling experiments in the same figure panel). Statistics have now been displayed on the graphs in Figure 1D for further clarification.

11. "Including EME with HAR prevented decay of the RPM signal, as predicted, since EME blocks elongation while enabling (even enhancing) puromycylation21,26." I find this very confusing. I understand that emetine blocks protein elongation whilst enabling puromycilation, but why does it block the effect of the protein initiation inhibitor Harringtonin? Do they compete with each other?

When ribosomes are frozen with emetine, they cannot transit mRNA and “fall off”. Therefore, the inclusion of EME in these experiments is a control to ensure that we are looking at true transit and runoff of ribosomes with harringtonine treatment (explanation in the second paragraph of “Flow RPM measures ribosome elongation rates in live cells” section)

12. Can the authors explain why the RPM signal of activated OT-I cells (PMA/Iono) increases 20fold compared to resting cells, but there is only a ~2-fold increase in signal in human cells? The authors previously mentioned: "We noted that the RPM signal in activated cells was only 2- to 5fold higher than in non-activated cells. This increase is modest compared to the ~15-fold activation-induced increase in protein synthesis in original studies 10,11. To examine this discrepancy, we incubated cells for 15 min with harringtonin (HAR) or pactamycin (PA) to block translation initiation or emetine (EME) or cycloheximide (CHX) to block elongation." Would the authors have followed the same path if they had started the paper with OT-I cells?

Human cells are not as well activated as OT-I in our study. The last question is beyond the scope of our reasoning as empirical evidence-based scientists, but we have applied for funding from the HG Wells Foundation for a time machine to answer this question.

13. Authors should include representative raw data of the flow cytometries used to perform the "Ribosome Transit Assay (RTA) in Figures 2 and 3 as supplemental data.

Done; now included in Supplemental Figures 1 and 3.

14. It would be interesting to compare RPM in T cells activated with a more physiological stimulus, such as beads anti-CD3 anti-CD28 vs PMA/Iono. Particularly after showing that peptide-specific stimulation (with SIINFEKL) is more effective than PMA/Iono in activating OT-I cells and inducing polysome formation (Figures 5B and Suppl Figure 4A).

We tried plate bound anti- CD3 and anti-CD28 early in these studies, and they didn’t induce as much early activation.

15. Can the authors include the gating strategy to call "activated OT-I cells" to the cells shown in Suppl Figure 3c?

A new Supplemental Figure 3D has been added showing the exact gating strategy for the OT-I cell RTA assays described in Supplemental Figure 3C and elsewhere.

16. In Figure 6B, the authors mention an increase in the volume of the cells based on the assumption of spherical morphology but then show an increase in diameter. It would be more consistent to show both parameters in the same graph.

The graph was changed to volume calculations instead of diameter for clarity. But they are linked as volume scales by radius cubed.

17. The paper's main conclusion (i.e., that the ratio of proteins to ribosomes in T cells activated in-vivo does not support their doubling time) is exciting. They conclude this after measuring cell volume, protein abundance, and ribosomes per cell. As no changes in cell volume and protein abundance between T cells activated in vitro vs in vivo were observed (Figures 6B and 6C), the difference is exclusively attributable to a reduced number of ribosomes per cell in T cells activated in vivo (Figure 6F). Critically, the measurement of ribosomes per cell in T cells activated in vivo (Figure 6F, "ex vivo day 2") includes only two data points. It is hard to understand how the authors calculated this figure's means and standard deviations as it is not described in the figure legend. From the dispersion observed for "day 1" and "day 2" in vitroactivated T cells, it seems that the variability of the assay to measure ribosome content could explain part of the phenotype. Additionally, there are several missing data points in Figure 6H.As this figure is just a transformation of Figures 6D and 6G, it isn't easy to understand why. Can I suggest that they include more data points for Figures 6F, G, and H in the ex vivo day 2' category as the two data points shown with little variability is out of keeping with the rest of the data, and may be skewing their data?

Figure 6F does not have the same number of data points as other panels because it required measurement of both protein content and ribosome number. Since the ribosome quantification method described here was developed later than some of our earlier protein measurements, not all experiments had both sets of data to properly calculate the proteins per ribosome. All data that had both values are included, though.

**Reviewer #3 (Recommendations For The Authors):**
Minor points:If an increase in cell diameter is recorded upon activation, why not also provide the value for the increase in volume?

Done

Regarding the writing, the erratic punctuation/hyphenation - or lack thereof - doesn't improve readability. One example: "....consistent with the idea that the flow RPM signal in day 1 resting lymphocytes...." Perhaps better: "... consistent with the idea that the RPM signal, obtained by flow cytometry for lymphocytes analyzed on day 1 and maintained in the absence of any activating agent,..." I understand that this can make for longer sentences, but I object to the use of 'flow' as shorthand for 'flow cytometry', and to the use of day 1 as an adverb or adjective. That works as lab jargon, it's less effective in a written text. The abbreviation 'DRiPs' is not defined. Words like 'notably', and 'surprisingly' can be eliminated.This work would benefit from the inclusion of a section describing 'Limitations of the study'.

This is now expanded in the Discussion, as described above.